# Monolithic three-dimensional integration of RRAM-based hybrid memory architecture for one-shot learning

Yijun Li[1], Jianshi Tang [1,2] ✉, Bin Gao [1,2], Jian Yao[3], Anjunyi Fan[4,5], Bonan Yan [4,5], Yuchao Yang [4,5,6,7], Yue Xi[1], Yuankun Li[1], Jiaming Li[1], Wen Sun[1], Yiwei Du[1], Zhengwu Liu [1], Qingtian Zhang [1,2], Song Qiu [3], Qingwen Li[3], He Qian[1,2] & Huaqiang Wu [1,2] ✉

In this work, we report the monolithic three-dimensional integration (M3D) of hybrid memory architecture based on resistive random-access memory (RRAM), named M3D-LIME. The chip featured three key functional layers: the first was Si complementary metal-oxide-semiconductor (CMOS) for control logic; the second was computing-in-memory (CIM) layer with HfAlO$_x$-based analog RRAM array to implement neural networks for feature extractions; the third was on-chip buffer and ternary content-addressable memory (TCAM) array for template storing and matching, based on Ta$_2$O$_5$-based binary RRAM and carbon nanotube field-effect transistor (CNTFET). Extensive structural analysis along with array-level electrical measurements and functional demonstrations on the CIM and TCAM arrays was performed. The M3D-LIME chip was further used to implement one-shot learning, where ~96% accuracy was achieved on the Omniglot dataset while exhibiting 18.3× higher energy efficiency than graphics processing unit (GPU). This work demonstrates the tremendous potential of M3D-LIME with RRAM-based hybrid memory architecture for future data-centric applications.

Artificial intelligence (AI) has made tremendous success during the past decade, driven by explosively big data and deep learning algorithms as well as rapidly evolving GPU and application-specific integrated circuit (ASIC) chips. However, a huge gap still exists between state-of-the-art AI hardware and the human brain, especially in terms of energy efficiency[1]. There are several key bottlenecks faced by the conventional computing hardware based on Si CMOS and von Neumann architecture with separated memory and computing units. For example, the slowing down of Moore's law scaling hinders the continuous improvement of integration density and performance of individual two-dimensional (2D) chips[2]. Moreover, the long latency of

memory access and limited data bandwidth between memory and computing units has become a key limiting factor of system performance, known as the von Neumann bottleneck[3]. To address these challenges, various technologies have been proposed from materials and devices to chip integrations and architectures, such as low-dimensional material transistors[4–6], emerging computational memory[7–9], heterogeneous and 3D integration[10–12], and memory-centric computing architecture[13–15], etc.

Among them, M3D emerges as an appealing technology that can potentially combine all these advances together to continuously increase the integration density, enrich the functionality and boost the

[1]School of Integrated Circuits, Tsinghua University, Beijing, China. [2]Beijing Advanced Innovation Center for Integrated Circuits, Tsinghua University, Beijing, China. [3]Suzhou Institute of Nano-Tech and Nano-Bionics, Chinese Academy of Science, Suzhou, China. [4]Institute for Artificial Intelligence, Peking University, Beijing, China. [5]Beijing Advanced Innovation Center for Integrated Circuits, School of Integrated Circuits, Peking University, Beijing, China. [6]School of Electronic and Computer Engineering, Peking University, Shenzhen, China. [7]Center for Brain Inspired Intelligence, Chinese Institute for Brain Research (CIBR), Beijing, China. ✉e-mail: jtang@tsinghua.edu.cn; wuhq@tsinghua.edu.cn

performance of a single chip[16]. Using back-end-of-line (BEOL)-compatible fabrication processes, M3D monolithically integrates multiple layers of different functions which are connected through fine-grain and dense vertical inter-layer vias (ILVs) to enable high-bandwidth data transfer between different layers[12]. System-level benchmarks suggest that M3D chips could exhibit 1000 times advantage in the energy-delay product over 2D counterparts for data-abundant applications like AI[17]. Recently, several prototype M3D chips have been reported with appealing performance. For example, M3D has been used to implement near-memory computing architecture[12] and computing-in-memory (CIM) architecture[18,19], exhibiting low latency, low power and enhanced integration density. To build a fully functional 3D system-on-chip (3DSoC) that can handle complex AI computing tasks, a monolithically integrated hybrid memory architecture is required, where different types of memories, such as high-efficiency CIM array, high-speed on-chip data buffer, and massively parallel TCAM, need to be vertically stacked and work synergistically. For example, one-shot/few-shot learning, which is a bio-plausible machine learning algorithm with minimum training cost[20–22], can be efficiently implemented by a memory-augmented neural network (MANN) with such hybrid memory architecture. Key features can be extracted using CIM-based convolutional neural network (CNN), then stored and retrieved from TCAM for classification[23–25].

To build such an M3D chip with hybrid memory architecture, RRAM and CNTFET are promising candidates thanks to their excellent BEOL compatibility and outstanding electrical performance. As a promising emerging memory, RRAM was first developed as digital memory for data storage with merits of simple structure, high integration density, fast speed, and low energy consumption[26–28]. It can achieve a high on/off ratio between the high-resistance state (HRS) and low-resistance state (LRS), which has also been used to efficiently implement TCAM[29–31]. TCAM can not only store data, but also compare them with a given input vector in parallel, which can be used for hardware search lookups and further realizing Hamming distance calculation[30,32,33]. Furthermore, analog RRAM with gradual resistive switching characteristics has also been developed for CIM to accelerate matrix-vector multiplication (MVM), which is the core and most computation-extensive operation in artificial neural networks (ANNs)[31,34–37]. By doing analog computing in situ via fundamental physics laws and eliminating data transfer, analog RRAM array could

achieve extremely high energy efficiency[37]. In addition, for the access transistors of RRAM and BEOL logic in M3D, CNTFET is a superior candidate with high mobility and low-temperature substrate-agnostic fabrication process[38–41]. It has been demonstrated that high-speed CMOS digital circuits and microprocessors can be built with CNTFETs, which can potentially provide nearly ten times performance improvement over Si counterparts[42,43]. With these memory and logic device elements, it remains challenging to design and fabricate an M3D chip of the desired hybrid memory architecture under the restrictions of thermal budget and process compatibility.

In this work, we report an M3D chip of RRAM-based hybrid memory architecture for efficient implementation of one-shot learning, where the RRAM-based CIM, TCAM and buffer arrays serve different functions as illustrated in Fig. 1a. The analog RRAM-based CIM arrays with the help of digital RRAM buffer were employed to implement CNN for feature extraction, while the digital RRAM-based TCAM arrays were used for template storage and matching. The fabricated M3D chip, named M3D-LIME, consisted of three key functional layers heterogeneously integrated. The first layer of Si CMOS was fabricated using a standard 130 nm CMOS process in the foundry, acting as control logic and data interface. The second layer of CIM array in the form of one-transistor-one-resistor (1T1R) was fabricated with HfAlO$_x$-based analog RRAM and Si access transistors underneath to implement MVM calculations of CNN for feature extraction. The third layer of 1T1R buffer and two-transistor-two-resistor (2T2R) TCAM arrays was fabricated with Ta$_2$O$_5$-based digital RRAM and CNTFET to implement template storing and matching. The on-chip buffer is critical to facilitate the data flow for the computations in the CIM layer underneath, which requires buffer to store intermediate results. The materials and devices were carefully optimized for each functional layer, and the last two layers were fabricated using a BEOL-compatible process at a low temperature (≤300 °C). Figure 1b shows the cross-sectional transmission electron microscope (TEM) image of the fabricated M3D chip to confirm its structural integrity. Here the ILVs between the 2$^{nd}$ and 3$^{rd}$ layers are not shown, and they can be fabricated using a standard BEOL interconnect process. As to be shown in the following sections, array-level electrical measurements and functional demonstrations on the CIM and TCAM arrays were carried out to verify the proper function of each layer. Furthermore, one-shot/few-shot learning was successfully implemented on the M3D-LIME chip to evaluate the performance

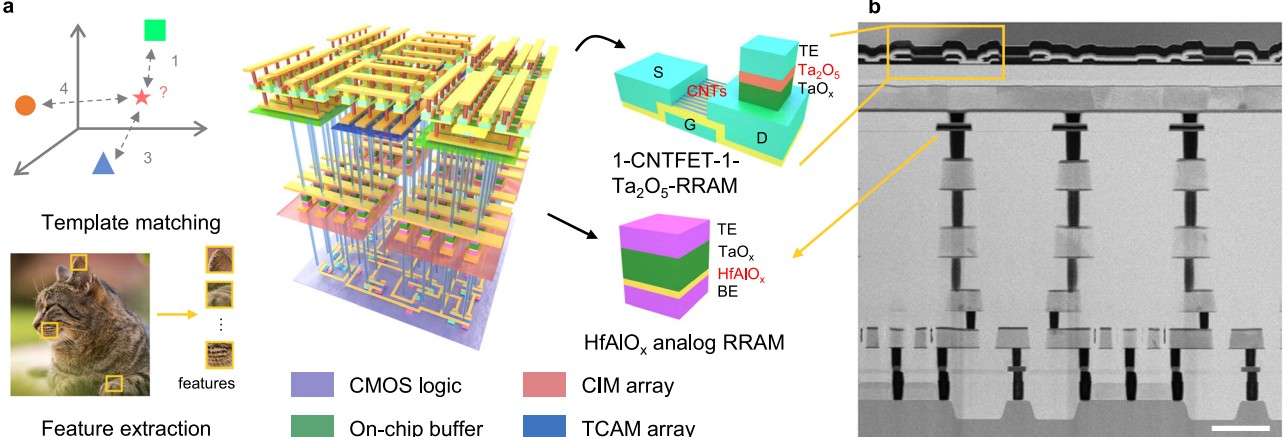

**Fig. 1 | M3D-LIME chip with hybrid memory architecture. a** Architecture of the M3D-LIME chip, which consists of three sequentially integrated layers. The 1$^{st}$ layer of Si CMOS logic is fabricated using a standard 130 nm CMOS process for control logic and data interface. The 2$^{nd}$ layer of CIM is fabricated with HfAlO$_x$-based analog RRAM 1T1R arrays to perform MVM calculations of CNN for feature extraction. The 3$^{rd}$ layer of TCAM is fabricated with CNTFET and Ta$_2$O$_5$-based digital RRAM in the form of 2T2R arrays to implement template storing and matching as well as in the form of 1T1R on-chip buffer for the CIM layer. Both the 2$^{nd}$ and 3$^{rd}$ layers are

fabricated using a carefully optimized BEOL-compatible process at a low temperature (≤300 °C) without affecting the pre-fabricated layers underneath. The left panels illustrate the implementation of one-shot/few-shot learning on the M3D-LIME chip. With the help of CMOS logic, feature extraction is implemented by the CIM and buffer arrays while template matching is performed on the TCAM array. **b** Cross-sectional TEM image of the fabricated M3D-LIME chip in this work, confirming the structural integrity of all three functional layers. Scale bar: 1 μm.

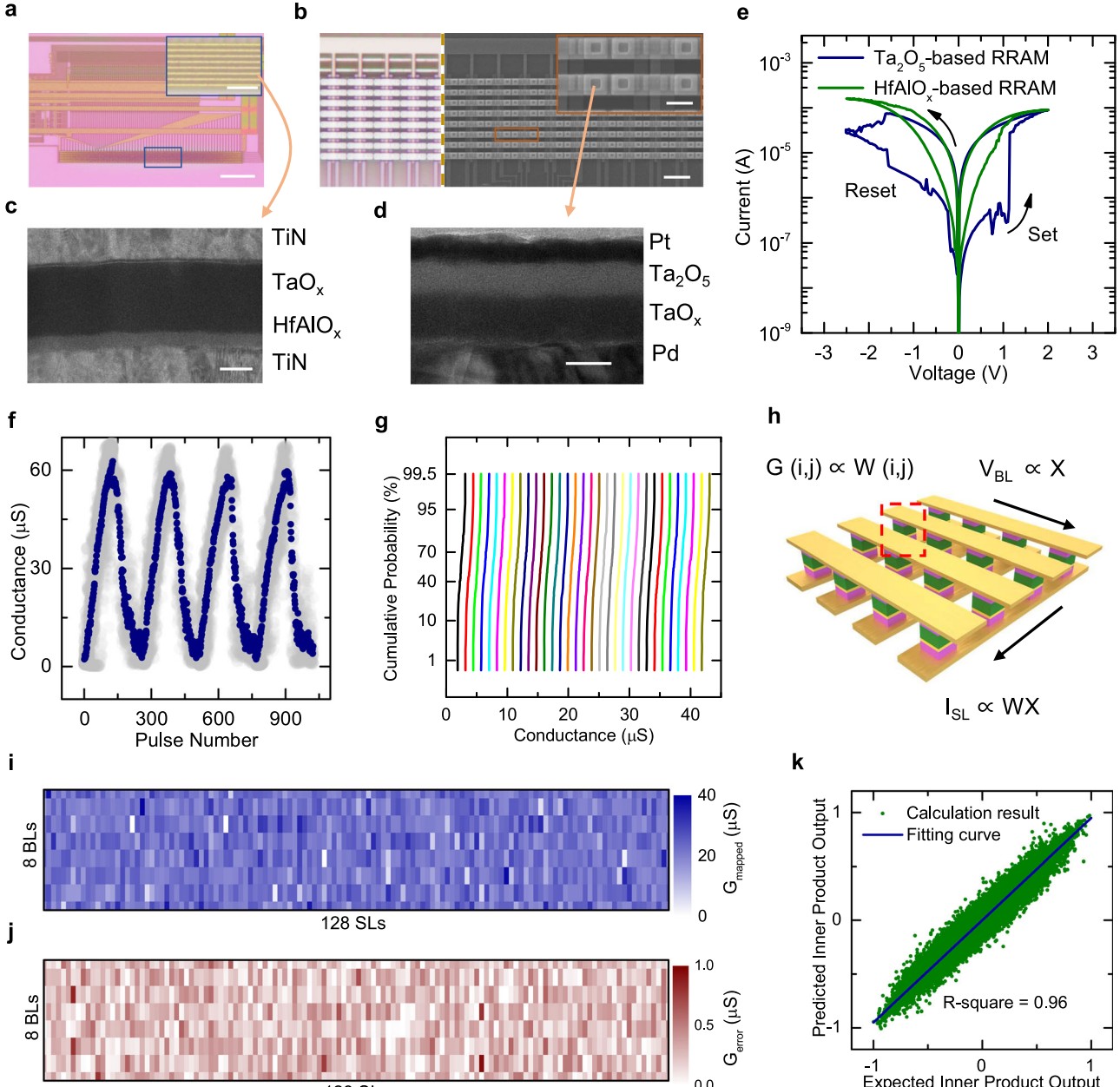

**Fig. 2 | Characterizations of analog RRAM-based CIM layer with M3D stacked RRAM buffer. a** Optical image of the 2nd layer of CIM array before the fabrication of the 3rd layer. Scale bar: 60 μm. Inset: zoom-in view of the CIM array, scale bar: 10 μm. **b** Optical image (left) and SEM image (right) of the 2nd layer after the fabrication of the 3rd layer. Scale bar: 5 μm. Inset: zoom-in view of the fabricated RRAM buffer array, scale bar: 1 μm. **c** and **d** Cross-sectional TEM images of the HfAlO$_x$-based analog RRAM and Ta$_2$O$_5$-based digital RRAM, respectively, scale bar: 20 nm. **e** DC I-V measurements of HfAlO$_x$-based RRAM with analog resistive switching characteristics and Ta$_2$O$_5$-based digital RRAM with a large HRS/LRS ratio. **f** Analog resistive switching characteristics of 20 1T1R cells in the CIM array under a series of set and reset pulses. Gray lines are the raw data, and blue line plots the average conductance of 20 devices measured. **g** Cumulative probability distribution of the CIM array with 32 equally distributed conductance states, showing the programming capability of 5 bits per cell. **h** Illustration of performing MVM operation on the CIM array. (**i**) Mapped conductance G$_{mapped}$ and (**j**) corresponding error G$_{error}$ after mapping a matrix on the 1k-bit CIM array. **k** Calculation result of MVM using the mapped array shown in (**i**) versus the theoretically expected result. The blue line plots the linear fitting with a small R-square of 0.96.

improvement brought by the M3D hybrid memory architecture. System-level simulations indicated that the M3D-LIME achieved a GPU-equivalent accuracy of up to 96% on the Omniglot dataset[22], while exhibiting 18.3×higher energy efficiency than GPU and 2.73× faster speed than its 2D counterpart.

## Results
### CIM layer with analog HfAlO$_x$-based RRAM array
On top of the Si CMOS logic layer, 1T1R analog RRAM array with a material stack of TiN/HfAlO$_x$/TaO$_x$/TiN was fabricated in the CIM layer

to implement CNN for feature extraction. Here 8 nm-thick HfAlO$_x$ served as the resistive switching layer (RSL) and 45 nm-thick TaO$_x$ served as the thermal enhanced layer (TEL) to enhance the analog switching characteristics[44–46]. The detailed fabrication process is elaborated in the Methods section. Figure 2a shows an optical microscope image of the 1T1R analog RRAM array in the 2nd layer of CIM before the fabrication of the 3rd layer of TCAM. In this image, the 1st layer of Si CMOS logic is also partially visible and acts as peripheral circuitry for data interface and control logic, including the access transistors for the analog 1T1R cells. The array is 1k-bit in size, with 8 bit lines (BLs) for

inputs and 128 source lines (SLs) for output. The BLs are visible in the optical image, as shown in the zoom-in view in the inset of Fig. 2a. On top of this analog RRAM array, a $Ta_2O_5$-based 512-bit digital 1T1R array was fabricated along with the 3rd layer of TCAM (2T2R array) using CNTFETs as the access transistors. Here the $Ta_2O_5$-based 1T1R array acts as an on-chip buffer for the CIM array and was optimized in the same way as the TCAM to be presented in the next section. An optical image of the 2nd layer of CIM array after the fabrication of the 3rd layer is shown in the left panel of Fig. 2b, and the corresponding scanning electron microscopy (SEM) image is shown in the right panel. The inset further displays the zoom-in SEM image of the $Ta_2O_5$-based 1T1R cells.

The two different types of RRAMs on the 2nd and 3rd layers were intentionally chosen to fulfill their functions as analog CIM and digital memory, respectively. To further verify the structural integrity, cross-sectional TEM analyses on the RRAMs in the last two layers were carried out after the fabrication of M3D-LIME chip. Figure 2c shows the TEM image of the analog RRAM in the CIM layer, in which the material stack of $TiN/HfAlO_x/TaO_x/TiN$ is clearly resolved. Figure 2d shows the cross-sectional TEM image of the digital RRAM in the TCAM layer, which has a different material stack of $Pd/TaO_x/Ta_2O_5/Pt$ to achieve a large on/off ratio of HRS/LRS. To reveal the different resistive switching characteristics of these two RRAMs, direct current (DC) I-V tests were carried out. Both RRAMs were tested in the form of 1T1R cell, and the results are shown in Fig. 2e. The $Ta_2O_5$-based RRAM exhibited an abrupt resistive switching in the set and reset processes with a large on/off ratio of HRS/LRS > 300, which is favorable for memory applications such as TCAM and buffer. Meanwhile, the $HfAlO_x$-based RRAM exhibited a more gradual resistive switching that enables excellent multi-level programming capability, which is considered favorable for CIM applications[44].

Furthermore, array-level electrical measurements were conducted on the fabricated 1k-bit analog RRAM array for CIM, and the detailed schematic of the array is shown in Supplementary Fig. 1. Figure 2f shows the analog resistive switching characteristics with good linearity and symmetry, where 20 1T1R cells were measured under a series of set and reset pulses with a width of 50 ns. The set, reset and read voltages are 1.6 V, 2.6 V and 0.2 V, respectively. Figure 2g shows the cumulative probability distribution of the CIM array with 32 equally distributed conductance states, equivalent to 5-bit programming precision. Here 128 1T1R cells in each state were programmed using the standard write-verify scheme, as shown in Supplementary Fig. 2. In addition, the results of array-level retention and endurance tests are shown in Supplementary Fig. 3, where excellent endurance over $10^6$ cycles and data retention exceeding $10^4$ s were achieved. These results confirm the superior analog switching characteristics of the $HfAlO_x$-based RRAM array for CIM.

Figure 2h illustrates the implementation of MVM on the array, where the weight matrix is first mapped onto the CIM array as the RRAM conductance and the vector is applied as the voltage pulse inputs to the BLs. By virtue of Ohm's law and Kirchhoff's current law, the current outputs on the SLs represent the MVM results[36]. To demonstrate this process on the fabricated 1k-bit CIM array, an 8×128 weight matrix used in the feature extraction later was first mapped on the CIM array as shown in Fig. 2i, and the corresponding mapping error is shown in Fig. 2j. The conductance values of RRAM cells were proportional to the magnitude of the elements in the weight matrix, which were quantized into 4 bits. After the weight mapping, a series of 8-bit vectors, whose elements followed Gaussian distributions, were input into the RRAM array for performing MVM operations. By applying the Kirchhoff's current law and Ohm's law, the RRAM array completed the MVM calculations and output a series of 128-bit vectors. All the elements of these vectors represented the inner products were normalized and plotted against their theoretically expected results in Fig. 2k. The linear fitting suggests a good consistency between them with a small R-square of 0.96. This result confirms the feasibility of

performing MVM operations efficiently on the analog RRAM array for CIM. The detailed experimental method is shown in Supplementary Note 1 and the characterizations on the CIM array in Fig. 2 were performed using on-chip control circuits (including WL address decoder and switch) with the assistance of an off-chip test system.

## TCAM array with digital $Ta_2O_5$-based RRAM and CNTFET

In the 3rd layer of M3D-LIME chip, CNTFET and $Ta_2O_5$-based digital RRAM were used to fabricate the 2T2R TCAM array, which performs template storage and matching for the one-shot/few-shot learning task. The fabrication process is well-optimized and elaborated in the Methods section, where one of the key challenges is to achieve high device performance within the thermal budget. The TEM structural analysis of the $Ta_2O_5$-based RRAM was already presented in Fig. 2. Figure 3a further shows a false-color SEM image of a $1 \times 5$ 2T2R TCAM array for electrical measurements. To build this array, the fabrication process of back-gated CNTFETs followed by $Ta_2O_5$-based RRAMs was carefully optimized under the highest temperature of 250 °C as elaborated in the Methods section. The fabricated CNTFET achieved an on-state current density $I_{on}/W$ up to 60 μA/μm, and the electrical data are shown in Supplementary Fig. 4. The high current density is favorable to drive the $Ta_2O_5$-based RRAM and achieve a high on/off ratio of HRS/LRS for memory applications. Before the fabrication of $Ta_2O_5$-based RRAM, the CNTFETs need to be carefully passivated to preserve their high performance. Here 10 nm-thick yttrium (Y) was first deposited on the CNTFETs and then naturally oxidized into $Y_2O_3$ to yield an excellent interface with CNT[47]. A bilayer oxide of 35 nm $Al_2O_3$/ 10 nm $HfO_2$ was then deposited by ALD at 250 °C to fully passivate the devices. The statistical results of 500 passivated CNTFETs were shown in Supplementary Fig. 5, showing an average $I_{on}/W$ of 28 μA/μm at $V_{DS} = -1$ V and a high $I_{on}/I_{off}$ ratio close to $10^5$.

After that, $Pd/TaO_x/Ta_2O_5/Pt$ digital RRAMs were fabricated on top of the drain terminals of CNTFETs to fulfill the 2T2R TCAM array. Here the sputtered 10 nm $Ta_2O_5$ and 20 nm $TaO_x$ served as the RSL and the oxygen reservoir layer, respectively, to achieve a large HRS/LRS ratio. After the etching of RRAM stack, 45 nm-thick $Al_2O_3$ was deposited by ALD at 150 °C to further passivate and protect the RRAM during operation. The fabricated $5 \times 1$ 2T2R TCAM array consists of 5-unit cells of 2T2R and a pre-charge transistor. Figure 3b shows its circuit diagram which employs CNTFETs to build the pre-charge circuitry, along with the illustration of using it for the calculation of Hamming distance. Here the sense amplifiers are not integrated on-chip in this study. The TEs of all the RRAMs are wired together as the match line (ML) which periodically charges and discharges controlled by the pre-charge transistor. Meanwhile, the drain electrodes of all the CNTFETs are wired together as the SL, while the gate electrodes of CNTFETs in a 2T2R pair are connected to the search lines (SEL and $\overline{SEL}$). The working principle for data storing and searching is illustrated in Supplementary Table 1. The template vector is stored in the TCAM while the extracted feature vector is input via SEL and $\overline{SEL}$ to calculate their Hamming distance. The discharging resistance of ML is decided by the number of mismatched bits ($N_{mis}$), giving rise to a time constant of discharging ($\tau$) inversely proportional to $N_{mis}$ ($\tau^{-1} \propto N_{mis}$). When calculating, the ML is first charged by the pre-charge transistor, and the extracted feature vector is then input. After turning off the pre-charge transistor and discharging the ML for a given time, the $N_{mis}$ that indicates the Hamming distance can be calculated by measuring the voltage of the ML.

The cross-sectional TEM image of a 1T1R half-cell in the TCAM layer is shown in Supplementary Fig. 6. Excellent endurance $> 5 \times 10^5$ cycles and retention exceeding $10^4$ s at 120 °C were also verified. Furthermore, the characterization of 2T2R TCAM cells in the TCAM layer shown in Supplementary Fig. 8 reveals a large resistance ratio >300× between the matched and mismatched cells, which is stable over a large number of search cycles ($>10^5$). To further demonstrate the proper function of the TCAM array, electrical measurement was

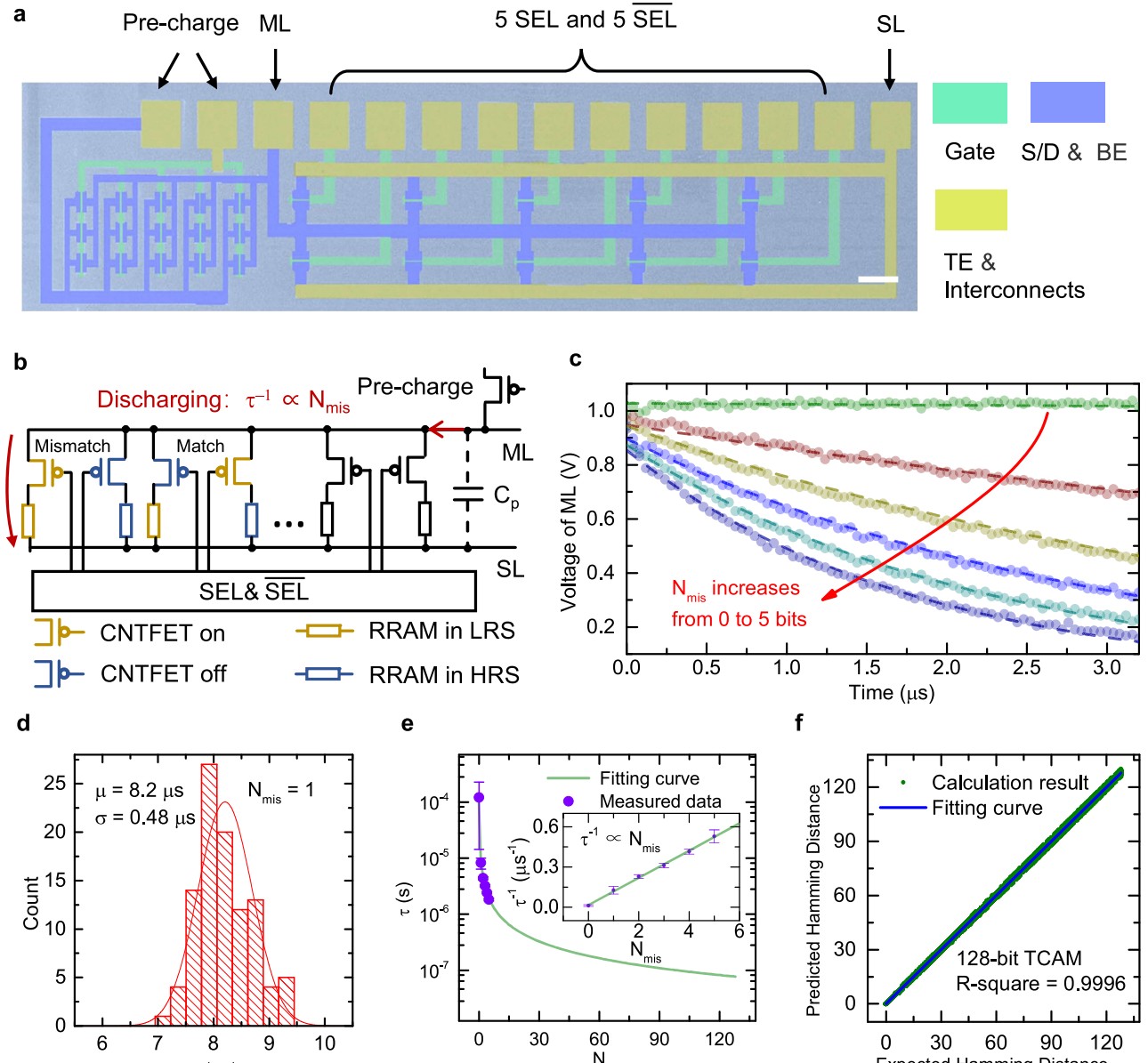

**Fig. 3 | Characterizations of digital RRAM-based TCAM array. a** False-color SEM image of a 1×5 2T2R TCAM array fabricated with CNTFET and Ta$_2$O$_5$-based digital RRAM. Scale bar: 50 μm. **b** Illustration of Hamming distance calculation using 2T2R TCAM. **c** The discharging waveform of ML as a function of the number of mismatched bits N$_{mis}$ (from 0 to 5). **d** Histogram of τ measured from 100 2T2R TCAM arrays with 1-bit mismatch (N$_{mis}$=1). **e** Fitting of the measured τ versus N$_{mis}$. The purple dots are the experimental data measured from 5 TCAM arrays. Inset: τ$^{-1}$ exhibits a linear dependence on N$_{mis}$. **f** Hamming distance (predicted) calculated by TCAM as a function of the actual Hamming distance (expected). The blue line plots the linear fitting with a small R-square of 0.9996, demonstrating proper function of Hamming distance calculation.

carried out to complete the search operation. As an example, a binary vector of "00000" was stored in the TCAM array by programming each 2T2R cell according to Supplementary Table 1. The source of the pre-charge transistor was fixed at a voltage of 1 V to initially charge the ML to 1 V by applying a voltage of -3 V to the gate. When calculating, 5 SEL and 5 $\overline{SEL}$ were applied with a voltage of 1 V or -3 V according to the input vector. After that, the pre-charge transistor was turned off by applying a gate voltage of 1 V, and then the ML started to discharge with a time constant depending on the N$_{mis}$ (τ ∝N$_{mis}^{-1}$). Figure 3c plots the discharging waveform measured by an oscilloscope. To measure the discharging waveform for different N$_{mis}$, the input vector was varied from "00000" to "11111", whose corresponding N$_{mis}$ increased from 0 to 5. Thanks to the optimized 2T2R array with a large resistance ratio between the matched and mismatched cells, the discharging

waveforms of different N$_{mis}$ were well-spaced, leading to an accurate calculation of the Hamming distance. The plot in Supplementary Fig. 9 confirms the inversely proportional relationship between τ and N$_{mis}$, which also verifies the proper search function of the RRAM-based TCAM. The detailed experimental method is described in Supplementary Note 2.

To further validate the proper function of template matching, multiple TCAM arrays were measured for statistical analysis. Figure 3d plots the histogram of τ for 100 2T2R TCAM arrays with 1-bit mismatch. For this measurement, the 2T2R cells in the arrays were mapped to store the vector "00000". After that, a randomly selected vector from "10000" to "00001" with N$_{mis}$ = 1 was input to each array. By measuring the discharging resistance (R$_{discharge}$) of each array with a read voltage of 0.15 V, τ can be quickly calculated by τ=R×C$_p$, where C$_p$ is the

parasitic capacitance. The statistical results indicate that the measured τ values exhibit a relatively narrow distribution, with an average of 8.2 μs and a standard deviation of 0.48 μs by Gaussian fitting. Furthermore, the discharging waveforms of TCAM arrays with an increasing $N_{mis}$ from 0 to 5 were also measured in the same way as in Fig. 3c to obtain a statistical result of τ. Figure 3e plots the extracted τ and $\tau^{-1}$ (inset), and the fitting curve confirms the linear dependence of $\tau^{-1}$ on the $N_{mis}$. Based on these experimental results, we can further simulate the calculation of Hamming distance for an even larger TCAM array with 128-bit vector inputs, as shown in Fig. 3f. The calculated Hamming distances by the TCAM array achieve excellent consistency with the theoretical values, as confirmed by the linear fitting with a small R-square of 0.9996. These results demonstrate that the fabricated $Ta_2O_5$-based digital RRAM arrays can successfully fulfill the desired TCAM operations.

### Implementation of One-Shot Learning

Based on the above measurement results of each functional layer, one-shot/few-shot learning was further implemented on the M3D-LIME chip to evaluate its system-level performance. Here a widely used MANN with the hybrid memory architecture is adopted, whose structure and key parameters are shown in Supplementary Fig. 10 and Supplementary Table 2. Using M3D-LIME, the CNN for feature extraction is implemented by the 2nd layer of CIM arrays with the help of buffer arrays in the 3rd TCAM layer and data interface in the 1st CMOS layer. Meanwhile, the templates are stored in the 3rd layer of TCAM arrays while template matching is performed by the massively parallel search operations. Figure 4a illustrates the data flow for implementing one-shot/few-shot learning by this MANN. In this task, massive data need to be frequently transferred among the three functional layers, which can take the advantage of high on-chip bandwidth enabled by the ultra-dense ILVs of M3D.

Figure 4b shows the detailed implementations of feature extraction and template matching using M3D-LIME. For the feature extraction, input data are first stored in the input buffer of the CIM array and then fed into the array via data interface, such as digital-to-analog converter (DAC). After that, the key operation of MVM is performed by the CIM array, and the output is read out by the data interface, such as analog-to-digital converter (ADC). Other operations such as max pooling and activation functions of ReLU can be implemented by the CMOS logic in the 1st layer using standard Si circuit design, as previously demonstrated[48–50]. The results are stored in the output buffer to complete the calculation of one layer in the CNN. This process can be repeated until the entire CNN is completed to extract features from a given image, which are then binary-quantized into binary feature vectors. For the template matching, binary feature vectors extracted from training are stored in the RRAM-based TCAM arrays, and then compared with a given input feature vector for inference. Here the Hamming distance is calculated via a large-scale parallel search operation on the TCAM as experimentally illustrated in Fig. 3. To classify a new query, the category with the minimum Hamming distance (i.e., the slowest discharging ML) is the final classification result.

Following the above workflow, the network is used to implement one-shot/few-shot learning on the Omniglot dataset and benchmark the system performance of M3D-LIME. As shown in Fig. 4c, accuracies of 89% and 96% can be achieved in the 5-way 1-shot and 5-shot learning, respectively. Both values are close to those obtained by GPU (93% and 98%). Furthermore, the execution time and energy consumption were also evaluated to manifest the advantages of M3D-LIME compared to GPU and 2D chip architecture which is illustrated in the Supplementary Fig. 11. The benchmark results in Figs. 4d and 4e show that M3D-LIME could achieve 18.3× higher energy efficiency than GPU (Nvidia Tesla V100 as a commonly used reference for benchmark[37, 51]), as well as a 2.73× speedup than its 2D counterpart. The detailed execution time and energy efficiency benchmarks are shown in Supplementary

Tables 3 and 4, respectively. The reported energy efficiency value of the Nvidia Tesla V100 GPU, which has been extensively utilized as a standard reference in numerous prior studies is included for comparison in the energy efficiency benchmark against GPUs. The pipeline implementation of MANN is also illustrated in Supplementary Fig. 12. For the 2D chip, the buffer is realized by a global cache nearby the CIM array, and data are transferred between the cache and the CIM array via a bus, where the bus bandwidth limits the number of CIM arrays computing in parallel and brings additional data transfer latency. By contrast, in the M3D-LIME chip, the buffer of each CIM array is realized by a local RRAM array located directly above it. As a result, data can be transferred more efficiently between the CIM array and the buffer directly via the high-density ILVs, which helps significantly reduce the latency.

## Discussion

In sum, we have designed and fabricated a M3D-LIME chip with a hybrid memory architecture of RRAM-based CIM, buffer, and TCAM for efficiently implementing one-shot learning. The chip consisted of three key functional layers, including a Si-based CMOS logic layer, a CIM layer with $HfAlO_x$-based analog RRAM array, and a TCAM layer with $Ta_2O_5$-based digital RRAM and CNTFET. RRAM-based on-chip buffer for CIM was also fabricated on the top layer to facilitate the data flow of CIM. The core devices were carefully selected for each functional layer and the fabrication processes were optimized to be compatible with BEOL integration. Extensive structural analysis and electrical measurements, including array-level CIM and TCAM demonstrations, were performed to verify the integrity and proper function of each layer. Excellent analog resistive switching characteristics with 5-bit precision and large on/off ratio >300× were achieved on the analog and digital RRAM arrays, respectively. Furthermore, the M3D-LIME chip was used to implement a MANN network for one-shot/few-shot learning on the Omniglot dataset, where high accuracy up to 96% was achieved. System-level benchmark further revealed that the M3D-LIME chip could achieve a 18.3× higher energy efficiency than GPU as well as a 2.73× speedup than its 2D counterpart. As illustrated in Supplementary Fig. 13, a scale-up M3D-LIME chip could monolithically integrate multiple CIM arrays, the associated on-chip buffers, and one or more TCAM arrays, on top of Si CMOS logic circuits to efficiently implement large-scale MANNs. Our work demonstrates the tremendous potential of M3D with hybrid memory architecture for future data-intensive AI and high-performance computing (HPC).

## Methods

### Preparation of high-quality CNT film

Preparation of high-purity (>99.99%) s-SWCNT solution: 10 mg of poly [9-(1-octylonoy)-9H-carbazole-2,7-diyl] (PCz) and 10 mg of arc-discharged single-walled carbon nanotubes (SWCNTs) were added into 20 mL of toluene. Then, the mixture was sonicated with an ultrasonic machine for 30 min. After that, the solution was centrifuged at 20,000 g for 1 hour to remove impurities, especially metallic SWCNTs (m-SWCNTs). Finally, the upper 95% of the supernatants were collected for further use.

Removal of dissociative PCz in solution: CNT solution was rinsed with tetrahydrofuran (THF) by the suction filter and the procedure was repeated for three times. Then, the CNTs on the filter membrane were re-dispersed into a chloroform dispersion. After that, the CNT/chloroform dispersion was diluted by ten times using toluene for subsequent wet transfer of s-SWCNT film onto the substrate.

Wet transfer and cleaning of the s-SWCNT film: A wafer substrate was put into the diluent for 48 hours, and a dense s-SWCNT film was then deposited on the wafer with a large amount of PCz on its surface. Then, excess PCz polymer was removed by rinsing the wafer in a toluene solution and adding 1.0% of trifluoroacetate (TFA) dropwise to the solution, leaving only a monolayer of PCz on the surface. For better removal of PCz, the mixture solution was heated at 80 °C for 1 hour.

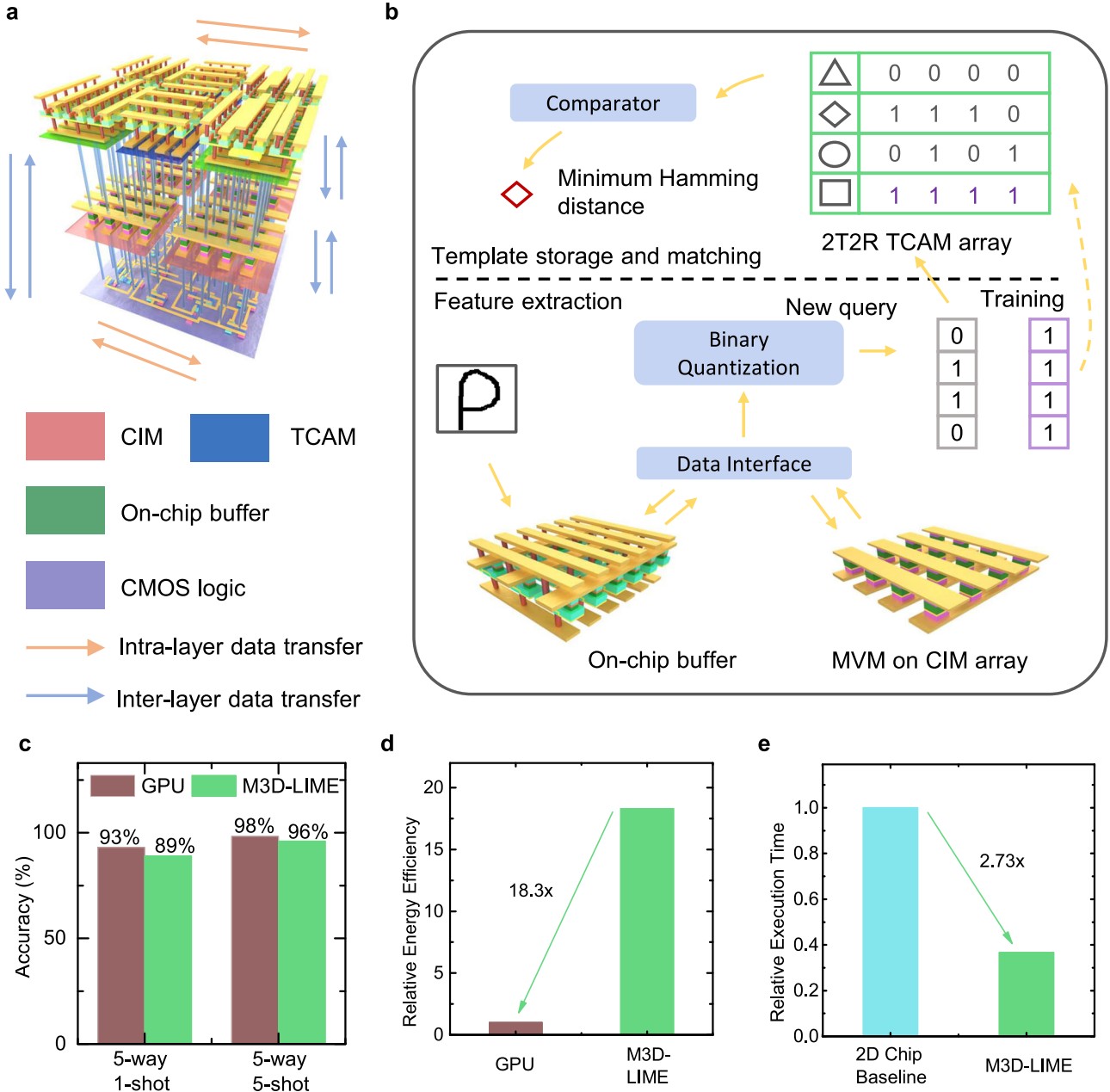

**Fig. 4 | Implementation of one-shot learning on M3D-LIME chip. a** Illustration of data flow on the M3D-LIME chip for implementing one-shot/few-shot learning. **b** Schematic of the implementation of one-shot/few-shot learning. It consists of two steps: For feature extraction, a CNN is implemented by the CIM arrays to perform MVM calculations with the help of the data interface in the 1st layer of Si CMOS logic and the 1T1R memory arrays for buffer in the 3rd layer; For template storage and matching, they are performed by the 3rd layer of TCAM. **c** The classification accuracy of one-shot/few-shot learning on the Omniglot dataset using GPU and the M3D-LIME. The accuracy is the average of 5 randomly selected classes (5-way) in the dataset. **d** Benchmark of the energy efficiency of the M3D-LIME chip and GPU. **e** Benchmark of the execution time on the M3D-LIME and 2D chip baseline.

## Fabrication of the M3D-LIME Chip

The 1st layer of CMOS logic was designed and fabricated on an 8-inch wafer, using a standard 130 nm Si CMOS process. The peripheral circuits of the CIM arrays were also fabricated in this layer, including the decoder, control logic and access transistors for the 1T1R cells. The wafer processing was stopped at the fourth metal layer (M4) with exposed W top vias after chemical mechanical polishing (CMP) for the fabrication of subsequent layers.

The 2nd layer of CIM was fabricated with HfAlO$_x$-based RRAM using a BEOL-compatible process. First, TiN with a thickness of 30 nm was sputtered as the bottom electrode (BE) of RRAM. Then, an 8-nm-thick HfAlO$_x$ was deposited as the RSL of RRAM using ALD at 300 °C.

After that, 45-nm-thick TaO$_x$ and 30-nm-thick TiN were sputtered as the TEL and TE of RRAM. The RRAM stack was then patterned by lithography and dry etch with Cl-based reactive ion etching (RIE). Next, 400 nm-thick SiO$_2$ was deposited by plasma-enhanced chemical vapor deposition (PECVD) at a temperature of 300 °C. The SiO$_2$ acts as the passivation layer to protect the fabricated RRAM stack, and it was patterned by lithography and dry etch with F-based RIE to form contact holes to the TE. Then, W vias were made to the TE, followed by surface planarization using CMP. After that, Al with a thickness of 500 nm was sputtered and then patterned using lithography and Cl-RIE to serve as BLs connected to the TE. Next, 100 nm-thick SiO$_2$ and 900 nm-thick Si$_3$N$_4$ were deposited by PECVD as the inter-layer

dielectric (ILD), and the surface was planarized again using CMP. Last, the ILD was patterned using lithography and Cl-RIE to form contact holes to the 3$^{rd}$ layer of TCAM. The highest temperature during the fabrication of the 2$^{nd}$ layer of CIM was 300 $^{o}$C, causing no damage to the prefabricated 1st layer of CMOS.

The 3$^{rd}$ layer of TCAM was then fabricated with CNTFET and Ta$_2$O$_5$-based digital RRAM, also using a BEOL-compatible process. Firstly, a stack of 10 nm-thick Ti, 15 nm-thick Pd and 1 nm-thick Ti was patterned as the gate metal of CNTFET by lithography and electron-beam evaporation (EBE). The high-work-function Pd was employed to ensure an appropriate threshold voltage of CNTFET, while the two Ti layers were to improve the adhesion. Then, HfO$_2$ with a thickness of 10 nm was deposited as the gate dielectric of CNTFET using ALD at 250 $^{o}$C. After that, HfO$_2$ was patterned using lithography and Cl-RIE to form contact holes to the gate metal. Then, high-purity CNT film was wet transferred and cleaned as the channel of CNTFET using the process described above. Next, a stack of 1 nm-thick Ti and 45 nm-thick Pd was patterned by lithography and EBE to serve as the source/drain contacts. Then, CNT film was patterned to define the channel of CNTFET by lithography and O$_2$ plasma etching. After that, a thin layer of Y$_2$O$_3$ was formed to optimize the interface between the CNT channel of and the passivation oxide later. This was done by first depositing Y with a thickness of 10 nm using EBE and baking the wafer in the air at a temperature of 250 $^{o}$C on a hot plane. Then, a passivation layer of 35 nm-thick Al$_2$O$_3$ and 10 nm-thick HfO$_2$ was deposited by ALD at 250 $^{o}$C. After that, contact holes were defined by lithography and dry etching using Cl-RIE. The Y$_2$O$_3$ layer, which was difficult to be dry etched, was then wet etched using hydrochloric acid (HCl). Next, a stack of 20 nm-thick TaO$_x$, 10 nm-thick Ta$_2$O$_5$ and 10 nm-thick Pt was sputtered for the digital RRAM, where Ta$_2$O$_5$ served as the RSL while TaO$_x$ acted as the reservoir of oxygen vacancy. Then the RRAM stack was dry etched using a 60 nm-thick Pd hard mask. Next, Al$_2$O$_3$ with a thickness of 45 nm was deposited as the passivation layer of RRAM by ALD at 150 $^{o}$C, followed by lithography and Cl-RIE to form contact holes. Last, 20 nm-thick Ti and 120 nm-thick Pd were patterned as the interconnects by lithography and EBE. The highest temperature during the fabrication of the 3$^{rd}$ layer of TCAM was 250 $^{o}$C, also causing no damage to the pre-fabricated two layers underneath.

## Data availability

Preliminary results from this study have been reported in the conference proceedings of the 2021 IEEE International Electron Devices Meeting (IEDM)[25]. The source data for Figs. 2–4 are provided in the Source Data file. Additional data supporting the findings of this study are available from the corresponding authors upon reasonable request. Source data are provided with this paper.

## Code availability

The code that supports the one-shot learning simulations in this study is available via GitHub at https://github.com/Tsinghua-LEMON-Lab/M3D_few_shot_learning. Other codes that support the findings of this study are available from the corresponding authors upon reasonable request.

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

## Acknowledgements

This work was in part supported by the STI 2030-Major Projects 2022ZD0210200 (J.T.), Natural Science Foundation of China 92264201 (J.T.) and 62025111 (H.W.), the XPLORER Prize (H.W.), the Tsinghua University Initiative Scientific Research Program (J.T.), and the Center of Nanofabrication, Tsinghua University.

## Author contributions

Y.J.L. and J.T. conceived and designed the experiments. Y.J.L., J.Y., Y.X. and W.S. performed the experiments with the help from B.G., S.Q., Q.L., H.Q. and H.W. A.F., J.L. and Y.D. contributed to the benchmark with the help from B.Y. and Y.Y. Y.K.L., Z.L. and Q.Z. contributed to the simulation. Y.J.L. and J.T. wrote the paper. All authors discussed and reviewed the manuscript.

## Competing interests

The authors declare no competing interests.
