## [Peer Review File · Nature Communications]

REVIEWER COMMENTS

Reviewer #1 (Remarks to the Author):

The authors present a new 3D integrated system on chip with CMOS logic, ReRAM based crossbar arrays for performing dot products, and ReRAM based TCAM for performing template matching and hamming distance calculation. This is a very interesting work, and authors performed a very good characterization. There is a rise of interest in TCAMs for in-memory computing, and this is the first time, to the best of my knowledge, that I see TCAM and crossbar integrated on the same chip. The paper is well written.

There are a few concerns that should be addressed before considering publications:

1. A schematic of the chip should be included. While Fig. 1 has a very nice 3D visualization, it is difficult to understand which circuits are in it. What components are integrated in the CMOS logic layer? Is the ADC on chip? Are all the required ReRAM programming circuit on chip?
2. An overview of the experiment setup would also help. How are data send to and read from the chip? It is complicated to understand which operation are performed on chip and which one are performed off chip.
3. Is the characterization of Fig 2 performed on the chip array or in external test structure? Please specify.
4. What is the size of the CNFET? That should be specified in order to understand if they are enough for forming/set/reset the ReRAM devices.
5. Is the pre-charge and sense amplifiers for the TCAM realized with CNFET or CMOS? Maybe this question can be answered with a clear schematic as pointed out in comment 1.
6. How did the experiment of SI Fig 5 was performed? How is the resistance measured?
7. Line 267 states that Fig. 3d shows a uniform distribution, but it looks Gaussian to me
8. Line 298 states that max pooling, ReLU, etc are implemented in the logic layer. What's the circuit for that?
9. How does this work compared with "Mao, R., Wen, B., Kazemi, A. et al. Experimentally validated memristive memory augmented neural network with efficient hashing and similarity search. Nat Commun 13, 6284 (2022). <https://doi.org/10.1038/s41467-022-33629-7>" ? I think the baseline for 2D implementation should be this paper
10. How was the GPU energy computed? Which model was consider?
11. How does a complete architecture look in the case of M3D-Lime?

Reviewer #2 (Remarks to the Author):

The manuscript demonstrates monolithic three-dimensional integration of hybrid memory architecture (M3D-Lime) based on resistive memories to implement one-shot learning.

The manuscript has a significant overlap with a previously published work (in IEDM 2021 from the same group).

The paper falls short in providing any novelty for this work and lacks clarity in its presentation of findings as well. The detailed comments can be found below. The reviewer recommends Rejection.

Comments:

The same work has been previously reported in [1], in IEDM, 2021, from the same group, which questions the novelty of this work.

In [1], the same M3D-LIME structure is demonstrated as proposed in the manuscript. All three layers, CMOS logic, RRAM-based CIM and CNTFET-based 2T2R TCAM

The application shown with M3D-Lime architecture is also the same, i.e., one-shot-learning.

The classification accuracy reported with the same structure in [1] is higher (97.8%) than what is reported in the present work (96%).

The improvement in energy efficiency for M3D-Lime when benchmarked with GPU, as reported in [1], is more (162x) compared to what is claimed in the present work (18.3x).

Moreover, [1] has not been referenced in the present manuscript.

Further, the paper cannot claim to be the first demonstration of the monolithic integration of hybrid memory architecture with resistive memories, as similar works have already been published.

In [2] demonstrates a NeuRRAM chip, a RRAM-based CIM chip, which integrates CMOS and memristors in a single architecture.

In [3], layers of computing, data storage, and input/output are fabricated on top of silicon logic circuitry in a three-dimensional architecture.

In [4], a monolithically integrated RRAM array with an oxide semiconductor channel access transistor in a 3D stack

Thirdly, while the paper claims to showcase a completely integrated system, it fails to demonstrate its complete performance in a comprehensive manner. Here, HfAlO_x-based RRAM is used for analog operations in CIM layer owing to the gradual resistive switching and multi-level programming capability. However, in dc-IV of HfAlO_x-based RRAM shown in Fig.2e demonstrate low MW (<10x). The paper lacks a clear explanation of how it achieves 32 levels with HfAlO_x, with such a low memory window (MW). Moreover, HfAlO_x-based memristors are filamentary and intrinsically stochastic in nature, exhibiting high variability. Although endurance and cumulative behavior are shown for 2-levels of conductance states, more multi-level endurance data is needed to determine whether the 32 distinct levels are maintained without disturbance.

The paper does not provide sufficient specifications for the GPU used, particularly in terms of speed, area, and energy consumption. The M3D-LIME and GPU are compared only based on the energy and accuracy, where the paper also reports low accuracy in comparison to GPU, which lowers the impact of work. How do the M3D-LIME area and speed (execution time) compare with the GPU?

The energy consumption of the CIM array is obtained using the XPEsim simulator (as mentioned in the Supplementary section). The parameters used for simulation are not reported. How do the energy values computed via simulations match the actual hardware?

The details of the 2D counterpart of the hybrid system used for execution time comparison with M3D-LIME are missing. The details are necessary to understand the execution time benchmark provided in the supplementary section of the paper. The reported values are not intuitive, and additional explanations should be provided.

The paper's Supplementary section only includes figures and captions, with no accompanying explanations or descriptions, making it difficult for readers to understand the study's results and implications.

References:

Y. Li et al., "Monolithic 3D Integration of Logic, Memory and Computing-In-Memory for One-Shot Learning," 2021 IEEE International Electron Devices Meeting (IEDM), San Francisco, CA, USA, 2021, pp. 21.5.1-21.5.4, doi: 10.1109/IEDM19574.2021.9720534.

Wan, W., Kubendran, R., Schaefer, C. et al. A compute-in-memory chip based on resistive random-access memory. *Nature* 608, 504–512 (2022). <https://doi.org/10.1038/s41586-022-04992-8>

Shulaker, M., Hills, G., Park, R. et al. Three-dimensional integration of nanotechnologies for computing and data storage on a single chip. *Nature* 547, 74–78 (2017). <https://doi.org/10.1038/nature22994>

J. Wu, F. Mo, T. Saraya, T. Hiramoto and M. Kobayashi, "A Monolithic 3D Integration of RRAM Array with Oxide Semiconductor FET for In-Memory Computing in Quantized Neural Network AI

Applications," 2020 IEEE Symposium on VLSI Technology, Honolulu, HI, USA, 2020, pp. 1-2, doi: 10.1109/VLSITechnology18217.2020.9265062.

Reviewer #3 (Remarks to the Author):

This is a very solid work in which the authors fabricate a monolithically 3D integrated small-sized chip that comprises a Si CMOS layer with two additional stacked BEOL-compatible layers respectively composed of analog RRAM for CIM core computing and binary RRAM for buffering and TCAM purposes. The authors report noteworthy results in terms of functional integration of the different functionalities on the same chip. The work represents an advancement as compared to existing literature and, in general, the claims are well supported. The adopted methodology is solid and the data are well presented, with a good amount of details being outlined. The paper is well written and enjoyable to read. However, a general limitation of the work is that the performance results of the classification task are just simulated since the authors fabricated a small-sized version of the chip having a single 1 by 5 TCAM array. In this respect, extra care must be taken when giving off numbers coming from the system-level simulations. While this reviewer understands that the fabrication of a full-sized chip is probably beyond the scope of this work, it is advisable to take care of some crucial aspects, outlined in the following.

- The most important point to be considered is that the delay and power consumption introduced by the data interface that, according to what the authors state includes DACs and ADCs (and possibly additional components and circuitry) is completely ignored. These elements can easily constitute the delay and energy bottleneck. This makes the estimated energy efficiency figure of merit inaccurate and the direct comparison with the GPU to be taken with a grain of salt, at best, or actually meaningless, at worst. A similar matter holds true for the latency benchmark vs. the 2D chip: if the delay ends up being dominated by the data interface, then the advantage of the M3D approach (at least in terms of speedup) vanishes. In this respect, either more accurate calculations needs to be done or at least these elements should be properly indicated in the paper.

- In supplementary table 4 the authors assume a 9 MOhm resistance for the mismatch case, which looks largely overestimated given the performance outlined in Fig. S5b (a value around 200 MOhm would be more representative). The authors should adopt a more realistic value, even if this does not change drastically the average write and read energies. Then, they assume that 1T1R buffer cells can be written in 50 ns with a single 3V (reset) / 1.5 V (set) pulse and read with a 0.15 V 10 ns pulse. However, this is not demonstrated in the paper - the authors are encouraged to show that achieving these performance levels is possible, providing a statistically significant validation. This is critical since the energy performance of the system largely depends on that of the buffer, given the numbers in supplementary table 4.

Additional remarks follows:

- Line 124: it would be ideal to provide a reference for the Ominglot dataset
- Line 176: please specify the set and reset pulse amplitude
- Line 181: it is advisable to show the full-fledged endurance plot of 1M cycles (i.e., through all cycles, not just using 1 point per decade)
- Line 191: please give more details as far as the normalized vector is concerned. How is this represented in Fig. 2k? How do the authors represent as a single number the results of a MVM operation? Does this actually average out possible significant errors in the individual entries of the output vector? Can the authors quantify the actual errors achieved at the single entry level?
- Line 239: similar to what stated previously, it is advisable to show the full-fledged endurance plot of 50k cycles. Also, in Fig. S4 it is clear that the memory window can go as low as 10. The authors are encouraged to show some statistical endurance and retention results on several devices.
- Line 241: in Fig. S5a the fact that there is no crosstalk is not evident to this reviewer. The authors should better convey their message perhaps using better data representation or a more dedicated approach. The legend in Fig. S5d can be misleading although the data are clear.

Response Letter to reviewers' Comments

We sincerely appreciate the reviewers' insightful and constructive comments. We have carried out additional experiments and revised the manuscript according to the reviewers' suggestions. Below are the point-by-point responses to each comment. All the changes to the manuscript are marked in **red**.

Reviewer #1

General comment:

The authors present a new 3D integrated system on chip with CMOS logic, ReRAM based crossbar arrays for performing dot products, and ReRAM based TCAM for performing template matching and hamming distance calculation. This is a very interesting work, and authors performed a very good characterization. There is a rise of interest in TCAMs for in-memory computing, and this is the first time, to the best of my knowledge, that I see TCAM and crossbar integrated on the same chip. The paper is well written. There are a few concerns that should be addressed before considering publications.

Response:

We would like to express our sincere gratitude to the reviewer for your time and efforts in reviewing the manuscript. The positive feedback and acknowledgment of the paper's novelty by the reviewer is truly encouraging. We highly value the insights and concerns raised by the reviewer regarding certain aspects of the manuscript. In response to these concerns, we have carefully revised the manuscript and have addressed each concern in a point-by-point manner in the response letter.

Comment#1:

A schematic of the chip should be included. While Fig. 1 has a very nice 3D visualization, it is difficult to understand which circuits are in it. What components are integrated in the CMOS logic layer? Is the ADC on chip? Are all the required ReRAM programming circuit on chip?

Response:

Thank you for the valuable feedback. We agree with the reviewer that a schematic diagram of the chip would help the understanding of the circuits and components integrated within it. In response to this comment, we have now added a detailed schematic diagram of the CIM array with Si CMOS circuits in the revised manuscript as **Supplementary Fig.1**. The schematic diagram of the TCAM array can also be found in the revised **Figure 3b**. Here the core peripheral circuits integrated in the CIM layer

include a 7:128 word line (WL) decoder and 128 WL drivers. While the ADC and programming circuits are not currently integrated on-chip, they can be readily realized using well-established circuit design in the Si CMOS logic layer underneath. In fact, a fully integrated CIM chip (as reported in Liu, et al., ISSCC, 500–502, 2020, doi: 10.1109/ISSCC19947.2020.9062953) that includes all these peripheral circuits is fabricated on the same wafer as the 1Kb RRAM array used in this work, as shown in **Figure R1**. we shall point out that the main focus of this work is to demonstrate a prototype M3D-LIME chip with hybrid memory architecture for one-shot learning, so only the core devices and circuits for each functional layer are integrated. The complete fabrication of a fully integrated M3D chip would require extensive engineering efforts, such as PDK development and complex circuit design, which is something we are still working on and beyond the scope of the current work.

Figure R1. Optical image of: a die in the fabricated wafer (a); the 1Kb RRAM array used in this work after the fabrication of the 3rd layer of TCAM (b) and before the fabrication of the 3rd layer of TCAM; the fully integrated CIM chip (as reported in Liu, et al., ISSCC, 500-502, 2020, doi: 10.1109/ISSCC19947.2020.9062953) after the fabrication of the 3rd layer of TCAM (d) and before the fabrication of the 3rd layer of TCAM (e). Scale bar: (a) 1 mm, (b) 6 μm , (c) 20 μm , (d) and (e) 200 μm .

Changes in the manuscript:

In page 7:

Furthermore, array-level electrical measurements were conducted on the fabricated 1k-bit analog RRAM array for CIM, and the detailed schematic of the array is shown in **Supplementary Fig.1**.

Add a new supplementary figure:

Supplementary Fig. 1. Schematic diagram of the 1Kb CIM array used for implementing MVM calculations. The peripheral circuits are implemented using Si CMOS logic in the first layer of M3D-LIME chip.

Revise Figure 3b:

Comment#2:

An overview of the experiment setup would also help. How are data sent to and read from the chip? It is complicated to understand which operation are performed on chip and which one are performed off chip.

Response:

Thank you very much for the kind suggestion. In our experimental setup, the data transfer and reading primarily relies on an off-chip testing system while assisted by on-chip control circuits. The off-chip testing system is responsible for importing data to the chip and reading computing results from it. Simultaneously, the on-chip control circuits help coordinate the data transfer and reading processes, ensuring accurate data input and output within the chip.

To further clarify the experimental setup, we have now included two supplementary notes, which explain the testing and demonstration methods depicted in **Figure 2** and **Figure 3** of the manuscript, respectively.

Changes in the manuscript:

In page 8:

The detailed experimental method is shown in **Supplementary Note 1**.

In page 10:

The detailed experimental method is described in **Supplementary Note 2**.

Add two new Supplementary notes:

Supplementary Note 1: Experimental method of results shown in Figure 2i-k.

In this experiment, we employed a multi-channel memory testing system in conjunction with the peripheral circuitry integrated on-chip with the CIM array (as depicted in **Supplementary Fig.1**) to demonstrate the calculation of MVM. The steps are summarized below:

To begin, we first mapped the weight matrix into the CIM array by programming. Initially, we determined the mapping relationship between the weights and the RRAM conductance values. For this demonstration, we set the memory window of the RRAM in the range of 0.4 to 40 μS , divided evenly into 16 conductance states (equivalent to 4 bits). Each weight in the weight matrix was quantized to 4 bits and then mapped to the respective RRAM conductance state, serving as the target value for programming the CIM array.

Subsequently, using the test system, we programmed each RRAM in the CIM array based on its target conductance using the standard write-verify method. We input a 7-bit binary address through the WL address input port of the CIM array to select the WL of the RRAM to be programmed. By controlling the voltage applied to the gate of the transistor connected to the selected WL through the WL reference voltage port, while keeping other WLs at a constant voltage of 0 V to ensure the connected transistors were turned off and prevent cross-talk, we could select one row of RRAMs (a total of 8) via WLs and then select the individual RRAMs one by one using the BLs (Bit Lines) orthogonal to the WLs.

For programming each 1T1R cell as shown in **Supplementary Fig.2**, we defined an interval based on the target conductance and the programming margin. The objective was to program the RRAM conductance within this interval, for which we applied a series of set and reset pulses. After each pulse, we read the conductance of the RRAM. If it exceeded the upper limit of the interval, we performed a reset operation in the next cycle. Conversely, if it fell below the lower limit, we performed a set operation in the next cycle. This process continued until the RRAM conductance was within the interval or the maximum number of cycles was reached, indicating the completion of programming. We then proceeded to program the next RRAM. The set and reset pulses had a width of 50 ns, and their voltages ranged from 1 V to 3 V, with WL reference voltages of 1 V to 3 V for set pulses and 5 V for reset pulses. For reading the conductance, we applied a WL reference voltage of 5 V and a BL voltage of 0.15 V.

Once all the RRAMs were programmed, we completed the mapping of the weight

matrix to the CIM array. By controlling the WL address input and BL, we read the results of the entire programmed array, as depicted in **Figure 2i**. The difference between this result and the target conductance value is shown in **Figure 2j**.

After that, we proceeded to input a series of vectors into the CIM array to demonstrate the MVM calculation. These input vectors followed a Gaussian distribution with a mean of 0 and a variance of 1, with each element of the vectors representing the elements of the matrix. These vectors were multiplied by the target conductance matrix, yielding the MVM result vector, where each element represented the expected inner product output.

Subsequently, by controlling the WL address and BL voltage, we performed 100 read operations on the CIM array. These operations were subject to read noise and write errors, resulting in fluctuating conductance matrices compared to the target conductance matrix. We multiplied the input vectors with these matrices, obtaining the predicted MVM results from the CIM array. In order to compare the difference between the predicted MVM results and the theoretical results, we considered the predicted inner product output as a function of its theoretical value, i.e., the expected inner product output. To mitigate the influence of output amplitude on the results, the data was normalized as plotted in **Figure 2k**, which demonstrates the capability of the CIM array in performing the MVM calculation.

Supplementary Note 2: Experimental method of results shown in Figure 3c.

Instruments utilized for this experiment included the Keysight B1500 and B1530 Semiconductor Analyzers, Keysight B2201 Switch Matrix, Tektronix MS054 Oscilloscope, and Probe Card System. To demonstrate the calculation of Hamming distance, we followed the steps outlined below:

Firstly, for each 2T2R cell in the TCAM arrays, we programmed all the RRAMs according to the scheme described in **Supplementary Table 1**. Specifically, we sequentially programmed each RRAM device by applying suitable voltages to the SELs, while turning off the driving transistors of the unselected RRAM (typical gate voltage of 0 V) and setting the gate voltage of the transistor driving the selected RRAM (typical gate voltage of -2.5 V). Next, we applied appropriate operating voltages to the ML and SL for DC programming of the devices. For the set operation, we applied 0V to the ML and -2 V to the SL. For the reset operation, we applied -3 V to the ML and 0V to the SL. After the set/reset operation, we then performed DC readout of the programmed RRAMs by applying -0.15 V to the ML, 0 V to the SL, -5 V to the SEL of the selected RRAM, and 0V to the SEL of the unselected RRAMs. If the resistance of the selected RRAM did not meet the requirements (typical values were around 30 k Ω for LRS and above 5 M Ω for HRS), we then applied a series of 50 ns pulses to continue programming the RRAM. After each 50 ns pulse operation, we read the resistance value of the RRAM. If it did not meet the requirements, we increased the pulse operating voltage for the next cycle until the resistance value fell within the target range to complete the programming. We repeated this process for each device until every unit in the 5 \times 1 TCAM array was correctly programmed. Through these operations, we successfully stored the template vector in the TCAM array. For the testing in Figure 3c,

a '00000' vector is stored.

After that, search vectors were inputted through the SELs to calculate the Hamming distance. In this experiment, we first connected the ML to the oscilloscope and set the source electrode of the pre-charging transistor to 1 V. Then, based on each bit of the search vector, we defined the on and off states of each transistor during the search process according to the scheme outlined in **Supplementary Table 1**. For the transistors in the off state, their connected SELs were grounded with SL, resulting in a gate voltage of 0 throughout the search process. For the SELs of transistors in the on state during the search process and the gate electrode of the pre-charging transistor, they were respectively connected to two pulse sources. Initially, the pre-charging transistor was turned on (typically with a gate voltage of -4 V) to charge the ML. After a certain period of time (typically 1 ms), the pre-charging transistor was turned off (typically with a gate voltage of 1 V), followed by a delay (typically 1 μ s). Then, by applying a -5 V pulse to the corresponding SELs, we turned on the corresponding transistors and triggered the oscilloscope to record the discharge waveform of the ML. The relevant pulses were applied through a set of PMUs of Keysight B1530, enabling good synchronization of the signals. By implementing search vectors with different Hamming distances from the stored template and recording the results on the waveforms, we obtained the results depicted in **Figure 3c**.

Add a new supplementary figure:

Supplementary Fig.2. Programming of analog 1T1R cells in the 2nd layer of CIM array. (a) Illustration of the overall programming process. (b) Illustration of the set operation. (c) Illustration of the reset operation.

Comment#3:

*Is the characterization of Fig 2 performed on the chip array or in external test structure?
Please specify.*

Response:

As mentioned in the earlier response, the relevant characterization was conducted using both on-chip and off-chip circuitry. Specifically, control circuits such as the WL address decoder, RRAM selector, and WL switch were implemented on-chip, while the ADC and write-read circuits were built in an off-chip test system. It is important to note that this measurement setup is primarily aimed at simplifying the demonstration and avoiding diversion from the main focus of this work, which is the M3D of a hybrid memory architecture for one-shot learning. As previously reported in our ISSCC paper (Liu, et al., ISSCC, 500–502, 2020, doi: 10.1109/ISSCC19947.2020.9062953), these circuits can be fully integrated on the chip and the corresponding chip is actually on the same wafer as the CIM array utilized in this work.

Changes in the manuscript:

In page 8: The linear fitting suggests a good consistency between them with a small R-square of 0.96. This result confirms the feasibility of performing MVM operations efficiently on the analog RRAM array for CIM. The detailed experimental method is shown in **Supplementary Note 1** and the characterizations on the CIM array in **Figure 2** were performed using on-chip control circuits (including WL address decoder and switch) with the assistance of an off-chip test system.

Comment#4:

What is the size of the CNFET? That should be specified in order to understand if they are enough for forming/set/reset the ReRAM devices.

Response:

We are grateful to the reviewer for this valuable suggestion. In response, we have made revisions to include further information on the channel size of CNFETs. Specifically, we utilized CNFETs with a channel size of $W/L=20\ \mu\text{m}/2\ \mu\text{m}$ to drive the RRAM. The choice of micron-scale channel size was limited by the available wafer-scale fabrication process in our university facility. Furthermore, CNFETs of this channel size have demonstrated the capacity to deliver an on-current of $\sim 500\ \mu\text{A}$ ($V_{ds} = -1\ \text{V}$, $V_{gs} = -5\ \text{V}$), which far exceeds the current requirement for driving Ta_2O_5 -based RRAM. Based on our previous experience, a drive current in the range of 100-150 μA is adequate for this purpose.

Changes in the manuscript:

- Supplementary Fig.5: **Supplementary Fig. 5. Statistical results of 500 CNTFETs.** (a) I_D - V_{GS} transfer curves of 500 measured CNTFETs with a drain voltage of -1 V. Histograms of (b) On-state current density, (c) subthreshold swing, (d) on/off ratio, and (e) field-effect carrier mobility. **The CNTFETs have a channel size of $W/L=20\ \mu\text{m}/2\ \mu\text{m}$.**

- Supplementary Fig. 6: **Supplementary Fig. 6. Characterizations of 1T1R half-cell in the TCAM layer.** (a) Cross-sectional TEM image of a 1T1R half-cell, scale bar: 200 nm. (b) Endurance test and (c) retention test (baking at 120 °C) of a typical 1T1R half-cell, exhibiting a large HRS/LRS ratio. (d) Histogram of set and reset voltages during the endurance test mentioned in (b). The endurance test involved the measurement of 10 1T1R half-cells, while the retention test included 25 1T1R half-cells with LRS and 25 1T1R half-cells with HRS. **The CNTFETs used to drive the Ta₂O₅-based RRAM cells have a channel size of W/L=20 μm/2 μm.**

- Supplementary Fig. 8: **Supplementary Fig. 8. Characterizations of 2T2R TCAM cells in the TCAM layer.** (a) DC I-V curves of two 1T1R half-cells in a 2T2R TCAM cell. (b) Distribution of discharging resistance for 100 2T2R cells, showing a large match/mismatch resistance ratio >300×. Results of the read disturb test are shown in (c) and (d): (c) Search ‘1’ when ‘1’ is stored in the TCAM cell. (d) Search ‘1’ when ‘0’ is stored in the TCAM cell. In the search operation, 100-ns pulses with a voltage of 1 V are applied to the TE (ML). **The CNTFETs used in the 2T2R TCAM cells have a channel size of 20 μm/2 μm.**

Comment#5:

Is the pre-charge and sense amplifiers for the TCAM realized with CNFET or CMOS? Maybe this question can be answered with a clear schematic as pointed out in comment 1.

Response:

As shown in **Figure 3a**, the pre-charge circuitry was implemented using CNTFET, but the sense amplifiers were not integrated on chip in this study. The relevant data were tested using external testing circuits. This M3D-LIME chip is a prototype that already includes many new devices, and adding sense amplifiers would make the testing process more complex. In the future work towards building a fully integrated M3D-LIME chip, the sense amplifiers could be designed and fabricated using mature Si CMOS technology. To avoid any confusion, we have clarified this point in the revised manuscript.

Figure 3a. False-color SEM image of a 1×5 2T2R TCAM array fabricated with CNTFET and Ta₂O₅-based digital RRAM. Scale bar: 50 μm .

Changes in the manuscript:

In page 9: ~~Figure 3b shows its circuit structure along with the illustration of using it for the calculation of Hamming distance.~~ **Figure 3b** shows its circuit diagram which employs CNTFETs to build the pre-charge circuitry, along with the illustration of using it for the calculation of Hamming distance. Here the sense amplifiers are not integrated on-chip in this study.

Comment#6:

How did the experiment of SI Fig 5 was performed? How is the resistance measured?

Response:

Thank you for the valuable comment. In response, we have added a supplementary note as follows that elucidates the testing method for **SI Fig 5** (now in revised **Supplementary Fig.8**).

Changes in the manuscript:

Add a new Supplementary note:

Supplementary Note 3: Testing method for Supplementary Fig.8.

We built a testing system with Keysight B1500 and B1530 for TCAM-related testing. Initially, we programmed the 2T2R TCAM cells using the configuration detailed in **Supplementary Table 1**. This involved applying distinct voltages to the gate terminals of the two CNTFETs to turn on the one driving the selected RRAM while keeping the other one driving the unselected RRAM turned off ($V_{GS}=0$). We also applied the appropriate voltages to the ML and SL in order to program each RRAM in the 2T2R cell. The typical voltages for set operation were $V_{GS}=-3$ V and $V_{ML}=-2$ V, whereas $V_{GS}=-5$ V and $V_{SL}=-3$ V for reset. Following the programming of the 2T2R cells, we conducted the 1-bit search operation on them using the configuration outlined in **Supplementary Table 1**. This included applying a voltage of 0.15 V to the ML and appropriate V_{GS} voltages to the two CNTFETs, then reading the current and calculating the resistance. In cases where the search data did not match the stored template, the CNTFET driving the RRAM in LRS opened and the CNTFET driving the RRAM in HRS closed, leading to a lower resistance known as mismatch resistance. Conversely, if the search data matched the stored template, the CNTFET driving the HRS opened and the CNTFET driving the LRS closed, resulting in a larger resistance known as match resistance. The ratio between the match and mismatch resistances serves as a key parameter for RRAM-based TCAM as it determines the length of the WL, or the search length.

Comment#7:

Line 267 states that Fig. 3d shows a uniform distribution, but it looks Gaussian to me.

Response:

Thanks for pointing out this for us. We intended to claim that the distribution plot indicates good uniformity in the measured τ rather than that τ has a uniform distribution. In our revised manuscript, we have corrected this statement.

Changes in the manuscript:

~~In page 10: The statistical result shows the measured τ has a relatively uniform distribution with an average value of 8.2 μs and a standard deviation of 0.48 μs .~~ **The statistical results indicate that the measured τ values exhibit a relatively narrow distribution, with an average of 8.2 μs and a standard deviation of 0.48 μs by Gaussian fitting.**

Comment #8:

Line 298 states that max pooling, ReLU, etc are implemented in the logic layer. What's the circuit for that?

Response:

Thanks for the comment. We did not actually integrate the circuits for max pooling and ReLU with our CIM array in this work. Nevertheless, as explained in the response to your Comment #1, the methods to design and fabricate these circuits using standard Si CMOS technology are well-established. The max pooling and ReLU function can be implemented using CMOS circuits, for example as shown in **Figure R2**. To avoid any confusion, we have clarified this point in the revised manuscript.

Figure R2. CMOS circuits implementation of max pooling (a) and ReLU function (b), as reported in Bin Zhao, et al., IECON, 2020, doi:

10.1109/IECON43393.2020.9254452 and Chao Geng, et al., MOCASST, 2020, doi: 10.1109/MOCASST49295.2020.9200299, respectively.

Changes in the manuscript:

In page 11: Other operations such as max pooling and activation functions of ReLU can be implemented by the CMOS logic in the 1st layer using standard Si circuit design, as previously demonstrated⁴⁸⁻⁵⁰.

In reference:

48. Liu, Q. *et al.* 33.2 A Fully Integrated Analog ReRAM Based 78.4TOPS/W Compute-In-Memory Chip with Fully Parallel MAC Computing. in *2020 IEEE International Solid- State Circuits Conference - (ISSCC)* 500–502 (IEEE, 2020). doi:10.1109/ISSCC19947.2020.9062953.

49. Geng, C., Sun, Q. & Nakatake, S. An Analog CMOS Implementation for Multi-layer Perceptron With ReLU Activation. in *2020 9th International Conference on Modern Circuits and Systems Technologies (MOCASST)* 1–6 (IEEE, 2020). doi:10.1109/MOCASST49295.2020.9200299.

50. Zhao, B., Chong, Y. S. & Tuan Do, A. Area and Energy Efficient 2D Max-Pooling For Convolutional Neural Network Hardware Accelerator. in *IECON 2020 The 46th Annual Conference of the IEEE Industrial Electronics Society* 423–427 (IEEE, 2020). doi:10.1109/IECON43393.2020.9254452.

Comment #9:

How does this work compared with “Mao, R., Wen, B., Kazemi, A. et al. Experimentally validated memristive memory augmented neural network with efficient hashing and similarity search. Nat Commun 13, 6284 (2022). https://doi.org/10.1038/s41467-022-33629-7”; ? I think the baseline for 2D implementation should be this paper.

Response:

Thank you very much for pointing out this important reference (Mao et al., Nat Commun 13, 6284, 2022), which is now included as reference [24] in the revised manuscript. This reference demonstrates that different structures of locality sensitive hashing (LSH) and TCAM in memory augmented neural network (MANNs) for one-shot learning can be efficiently implemented in a fully integrated memristive crossbar platform. However, it may not be an appropriate 2D baseline for the benchmark of M3D-LIME as the used transistor and RRAM resistance range are quite different. We shall point out that the selection of a “good” 2D baseline for M3D itself is quite challenging as the same task shall be realized with similar functional components. The primary objective of execution time benchmark in this work is to demonstrate the effectiveness of M3D technology in improving the processing speed of the chip through efficient communication using ILVs. In order to make a more reasonable comparison,

other conditions of the 2D baseline should be made as similar as possible to M3D-LIME, except relying on 2D bus communication. However, the device parameters, TCAM architecture, and scale of the neural network used in the referenced paper differ significantly from those employed in this work on M3D-LIME, as shown in **Table R1** below. Furthermore, since the reference did not report its computing latency, we are unable to calculate and compare it with the computing latency of M3D-LIME.

Comparison	This work	Ref. [24]
Chip structure	M3D integration	2D integration
TCAM cell	2-CNTFET-2-RRAM (2T2R)	2-RRAM (2R)
LRS of RRAM	$\sim 30\text{k}\Omega$ (TCAM), $\sim 25\text{k}\Omega$ (CIM)	$\sim 6\text{k}\Omega$
Network demo	CNN+TCAM	CNN+LSH+TCAM

Table R1. Comparison of this work and ref. [24] (Mao et al., Nat Commun 13, 6284, 2022).

Comment#10:

How was the GPU energy computed? Which model was consider?

Response:

Thanks for your comment. In this work, we did not directly compare the energy consumption of GPU and M3D-LIME chip as their technology nodes are vastly different. Instead, we compare the energy efficiency as shown in the **Supplementary Table 4**. Here we adopted the reported energy efficiency value (100 GOPS/W) of NVIDIA Tesla V100 GPU, which has been widely used as a standard reference in many prior works (e.g., S. Ambrogio et al., *Nature* 2018; P. Yao et al., *Nature*, 2020.) for energy efficiency benchmark against GPU. To avoid any confusion, we have clarified this point in the revised manuscript.

Changes in the manuscript:

In Supplementary Table 4: Energy consumption evaluation for (a) TCAM, (b) CIM arrays, (c) buffer and (d) M3D-LIME as well as comparison of energy efficiency with GPU. #These values are obtained by the simulator XPEsim (Wenqiang Zhang, et al., DAC, 2019, doi: 10.1145/3316781.3317797). &Peripheral CMOS circuits, including the ADCs, DACs, WL switch, and BL switch are all taken into consideration. *The reported energy efficiency value of the Nvidia Tesla V100 GPU (Stefano Ambrogio, et al., *Nature*, 2018, doi: 10.1038/s41586-018-0180-5), which has been extensively utilized as a standard reference in numerous prior studies is included for comparison in the energy efficiency benchmark against GPUs. The Nvidia Tesla V100 GPU is fabricated at 12 nm technology node.

Comment#11:

How does a complete architecture look in the case of M3D-Lime?

Response:

Thanks for your comment. A complete M3D-LIME architecture proposed in this work consists of three key functional layers heterogeneously integrated on a single chip. The first layer consists of high-speed Si CMOS logic circuits fabricated using standard Si CMOS technology. It is utilized for the peripheral circuits required for the CIM and TCAM arrays on the top layers, such as SA, ADC, and control circuits, as well as for the computing functions that are difficult to implement with RRAM and CNTFETs, such as ReLU and max pooling. The second layer features multiple CIM arrays built with HfO₂-based analog RRAM. With the assistance of the peripheral circuits implemented in the first layer of Si CMOS logic, the CIM arrays would perform the MVM computations and implement a CNN for feature extractions. The extracted features are then stored in an on-chip RRAM buffer stacked above it. The third layer consists of the on-chip 1T1R buffer array and a large-scale 2T2R TCAM array built with Ta₂O₅-based binary RRAM with a high ratio of HRS/LRS. The on-chip buffer array stacked above the CIM arrays stores the computing results from the CIM arrays and communicates through the fine-grain and high-density interlayer vias (ILVs), significantly reducing the data transfer delay. The TCAM array performs Hamming distance calculations with the assistance of the peripheral circuits implemented in the first layer of Si CMOS logic, and the communication between these two layers can also be accelerated through the high bandwidth ILVs. As a result, with these three functional layers, which heterogeneously integrate multiple devices and work synergistically, the M3D-LIME architecture can accomplish one-shot learning tasks efficiently.

I hope this answers your question.

Reviewer #2

General comment:

The manuscript demonstrates monolithic three-dimensional integration of hybrid memory architecture (M3D-Lime) based on resistive memories to implement one-shot learning. The manuscript has a significant overlap with a previously published work (in IEDM 2021 from the same group). The paper falls short in providing any novelty for this work and lacks clarity in its presentation of findings as well. The detailed comments can be found below. The reviewer recommends Rejection.

Response:

We would like to express our gratitude to the reviewer for dedicating their time and efforts to thoroughly reviewing our paper. The reviewer has raised questions regarding the novelty of our work, and we would like to provide further clarification as outlined below.

Firstly, it is important to note that the IEDM conference paper has a strict length limit of four pages. Due to this constraint, it is common for some details to be omitted. However, to ensure transparency and encourage the disclosure of experimental results, IEDM explicitly states that "**Publication in the digest in no way precludes later publication of a fuller account of the work in another journal**" (<https://www.ieee-iedm.org/electronic-submission>). Indeed, there are numerous examples where IEDM papers have been later expanded and published in journals to provide a more comprehensive account of the research.

Secondly, this paper is not a simple repetition of our previous work reported at IEDM. In fact, it has made significant improvements in multiple aspects, including architecture, fabrication, characterization, functional demonstration, and system evaluation, as summarized in the **Table R2** below. It is worth mentioning that **all the figures used in this paper are completely new, with no reuse of images in IEDM paper.**

Comparison		IEDM paper	This work
		TCAM + CIM + Si CMOS logic	TCAM + CIM + Buffer + Si CMOS logic
BEOL Transistor	Fabrication process	Introducing Y ₂ O ₃ for interface passivation and low-temperature wet cleaning for mobility improvement	
	Current density	~ 8 $\mu\text{A}/\mu\text{m}$	~ 30 $\mu\text{A}/\mu\text{m}$
Characterization of the 3 rd layer TCAM		Single devices	Statistic data from 2T2R TCAM array

Demonstration of the 2 nd layer CIM	Multi-bit programming capability	MVM calculation in the CIM array
Demonstration of the 3 rd layer TCAM	2T2R TCAM cell	Hamming distance calculation in the 2T2R TCAM array
Implementation of one-shot learning	Based on measurement data from single devices	Based on array-level measurement results
Execution time benchmark	Array-level analysis	Pipeline-level evaluation
Speedup compared to 2D chip baseline	1.68×	2.73×

Table R2. Comparison of IEDM paper and this work.

Here are the key improvements compared to our previous IEDM paper:

- (1) In terms of architecture, we further enriched the M3D-LIME architecture by **integrating an on-chip RRAM buffer in the 3rd layer of the TCAM and stacking it directly above the CIM array in the 2nd layer.** This hybrid memory architecture is driven by two primary factors. Firstly, for the one-shot learning task, the CIM array usually needs to be much larger than the TCAM array, which could result in a low area utilization of the 3rd layer. Secondly, subsequent benchmarks have shown that in traditional 2D architecture, which uses a bus to transfer data between the CIM arrays and caches, the efficiency of CIM array computation is mostly limited by the data transfer efficiency. By implementing an on-chip RRAM buffer stacked above the CIM arrays and utilizing fine-grain and ultrahigh-density inter-layer vias (ILVs) for efficient data transfer, we can effectively reduce the data transfer latency between CIM arrays. Consequently, this helps enhance the computational speed of the CIM system for neural networks, particularly for large-scale networks.
- (2) In terms of BEOL CNTFETs, the fabrication process has also been optimized to improve their device performance, especially the drive current, and reduce the operating voltage. In our work presented at the IEDM, we employed Al₂O₃ as the passivation layer to provide protection against potential damage induced by subsequent fabrication steps and also electrical isolation for the CNTFETs. To further enhance the interface between CNTs and oxide, **we introduced Y₂O₃ as a buffer layer** in this work. Additionally, we incorporated a wet cleaning step that was not employed in the IEDM paper. This step was specifically designed to eliminate polymer residues from the surfaces of CNTs for M3D fabrication, where high-temperature annealing needs to be avoided. These two process

optimizations successfully **increased the current density of CNTFETs from $\sim 8 \mu\text{A}/\mu\text{m}$ to $\sim 30 \mu\text{A}/\mu\text{m}$** , resulting in much enhanced driving capability. Moreover, we optimized the etching and atomic layer deposition (ALD) processes and **utilized a 10 nm HfO_2 gate oxide** as a substitute for the previously used 5 nm HfO_2 +10 nm Al_2O_3 gate oxide to reduce the operating voltage. These fabrication optimizations serve as the basis for device testing and functional demonstrations on a larger scale than what was originally reported at IEDM.

- (3) In terms of electrical characterizations, especially on the 3rd layer TCAM, further testing was conducted to demonstrate the feasibility of the proposed M3D architecture. In the IEDM paper, in order to verify the functionality of the TCAM fabricated in the 3rd layer, only single devices were characterized, confirming that Ta_2O_5 -based RRAM has a high HRS/LRS ratio and is suitable for implementing 2T2R TCAM. For the characterization of the 2T2R cell, only read disturb characteristics were evaluated. In this work, benefiting from the above optimizations of fabrication process, **a 5×1 TCAM array was fabricated for characterization**. The discharge waveforms were measured under different N_{mis} (the number of different bits in the query vector versus the storage vector), and the discharge time constants were calculated for further simulation. These tests further validate the feasibility of the M3D proposed architecture.
- (4) In terms of functional demonstration, this study comprehensively verified that each functional layer is working properly, which was not done in the IEDM paper. Beyond the analog switching characteristics and multi-bit programming capability of the 2nd layer CIM in the IEDM paper, in this study **we further demonstrated the implementation of MVM calculation using the CIM array**. To do so, we first mapped a weight matrix used in a real task to a 1k-bit CIM array, where the conductivity of the RRAM cells corresponded to the weight values. We then input a series of input vectors with elements following a Gaussian distribution. Next, MVM calculation was performed in situ using Ohm's law and Kirchhoff's current law. We then compared the MVM calculation results (predicted inner product output) with their theoretical values (expected inner product output) to show that the RRAM CIM array in the proposed M3D architecture can correctly implement MVM calculation. Regarding the 3rd layer of TCAM, the IEDM paper only reported test results of single devices to demonstrate the high HRS/LRS ratio of 1-CNTFET-1- Ta_2O_5 -RRAM cells for implementing 2T2R TCAM. In this study, **we further demonstrated the Hamming distance calculation using a fabricated 2T2R TCAM array**. Based on the test results, we statistically analyze the relationship between the discharge

time constant and N_{mis} and validate it through TCAM array test results. We then demonstrated the Hamming distance calculation for a 128-bit vector using the TCAM array. We compare the predicted hamming distance with the expected Hamming distance to show that the 2T2R TCAM array in the proposed M3D architecture can fulfill the template matching function. The experimental demonstration results of MVM calculation and Hamming distance calculation also allowed for a more accurate estimation of the accuracy of this architecture for one-shot learning tasks compared to the IEDM paper.

- (5) In terms of benchmarking energy consumption and execution time, several modifications were made. Firstly, **for the energy consumption benchmark, we conducted a more detailed analysis of the test results and provided as many details as possible in the results and intermediate steps.** It is important to note that in the IEDM paper, we scaled the M3D chip to 28 nm in order to make a fairer comparison with the GPU (NVidia TESLA V100 is based on the 16 nm technology node). However, such scaling is tricky as we did not actually fabricate the M3D chip at 28nm technology node. Therefore, in this paper, we based the energy consumption of the M3D chip on the actual 130 nm technology node for comparison. This is one reason why the energy efficiency advantage in this paper is significantly smaller than the one reported in the IEDM article. Another reason is the use of the Ta₂O₅-based RRAM array as a buffer, which has a slightly higher energy consumption compared to Si CMOS-based SRAM buffer in the 2D baseline. However, the advantage of the RRAM buffer array lies in the enhanced data transfer efficiency by stacking it above the CIM array in the M3D architecture, as well as its significantly smaller area compared to SRAM. Secondly, **for the execution time benchmark, we conducted a pipeline-level evaluation by taking into account the parallel processing in actual tasks.** It is important to mention that in the IEDM paper, the 2D baseline comparison was made between TCAM and CIM arrays on two separate chips. In this paper, however, we adopted a more reasonable approach for the 2D baseline, fabricating TCAM and CIM arrays on the same chip. The only difference between the 2D baseline and the M3D architecture is that the M3D architecture uses the ultrahigh-density inter-layer vias (ILVs) for data transfer between the CIM and TCAM arrays, while the 2D baseline uses on-chip data buses.

Overall, we sincerely appreciate the reviewer's comments and are grateful for the opportunity to address their concerns and provide additional context for our work. Our point-by-point responses to your technical comments are as follows.

Comment #1:

The same work has been previously reported in [1], in IEDM, 2021, from the same group, which questions the novelty of this work. In [1], the same M3D-LIME structure is demonstrated as proposed in the manuscript. All three layers, CMOS logic, RRAM-based CIM and CNTFET-based 2T2R TCAM. The application shown with M3D-Lime architecture is also the same, i.e., one-shot-learning. The classification accuracy reported with the same structure in [1] is higher (97.8%) than what is reported in the present work (96%). The improvement in energy efficiency for M3D-Lime when benchmarked with GPU, as reported in [1], is more (162x) compared to what is claimed in the present work (18.3x). Moreover, [1] has not been referenced in the present manuscript.

[1] Y. Li et al., "Monolithic 3D Integration of Logic, Memory and Computing-In-Memory for One-Shot Learning," 2021 IEEE International Electron Devices Meeting (IEDM), San Francisco, CA, USA, 2021, pp. 21.5.1-21.5.4, doi: 10.1109/IEDM19574.2021.9720534.

Response:

Thank you for the reviewer's kind suggestion. Sorry we forgot to cite the previously published IEDM paper in the paper, which was only explicitly mentioned in the cover letter. We have now included it as reference [25] in the revised manuscript. In response to the general comment, we have provided a thorough explanation in the above response to your general comment. On top of our previous IEDM paper, we further enriched the M3D-LIME architecture by integrating an on-chip RRAM buffer in the 3rd layer of the TCAM and stacking it directly above the CIM array in the 2nd layer. We further optimized the fabrication process and device performance in each layer, and built a prototype 5×1 2T2R TCAM array, which enabled us to systematically characterize the array-level performance of each building block in M3D-LIME. The new measurement data were then employed to perform a more comprehensive and accurate benchmarking. Therefore, even both works implemented the same one-shot learning task, the obtained accuracy and energy efficiency in the system-level benchmark could be different given their different chip architectures and experimental data.

Changes in the manuscript:

In page 18:

25. Li, Y. et al. Monolithic 3D Integration of Logic, Memory and Computing-In-Memory for One-Shot Learning. in 2021 IEEE International Electron Devices Meeting (IEDM) vols 2021-December 21.5.1-21.5.4 (IEEE, 2021).

Comment#2:

The paper cannot claim to be the first demonstration of the monolithic integration of hybrid memory architecture with resistive memories, as similar works have already

been published. In [2] demonstrates a NeuRRAM chip, a RRAM-based CIM chip, which integrates CMOS and memristors in a single architecture. In [3], layers of computing, data storage, and input/output are fabricated on top of silicon logic circuitry in a three-dimensional architecture. In [4], a monolithically integrated RRAM array with an oxide semiconductor channel access transistor in a 3D stack.

[2] Wan, W., Kubendran, R., Schaefer, C. et al. A compute-in-memory chip based on resistive random-access memory. *Nature* 608, 504–512 (2022). <https://doi.org/10.1038/s41586-022-04992-8>

[3] Shulaker, M., Hills, G., Park, R. et al. Three-dimensional integration of nanotechnologies for computing and data storage on a single chip. *Nature* 547, 74–78 (2017). <https://doi.org/10.1038/nature22994>

[4] J. Wu, F. Mo, T. Saraya, T. Hiramoto and M. Kobayashi, "A Monolithic 3D Integration of RRAM Array with Oxide Semiconductor FET for In-Memory Computing in Quantized Neural Network AI Applications," 2020 IEEE Symposium on VLSI Technology, Honolulu, HI, USA, 2020, pp. 1-2, doi: 10.1109/VLSITechnology18217.2020.9265062.

Response:

Thanks for the comment. We shall clarify that we never claimed that this work is the first demonstration of the monolithic integration of RRAM and Si CMOS, which has been reported in numerous prior arts as the reviewer pointed out. Rather, this work reports the first demonstration of monolithic three-dimensional integration (M3D) of a hybrid memory architecture that includes RRAM-based CIM, TCAM and buffer arrays serving different functions in the M3D-LIME chip. To the best of our knowledge, there have been no reports on M3D architectures integrating multiple types of RRAM. The CIM, TCAM, and buffer integrated in this work efficiently perform MVM calculation, Hamming distance calculation, and data storage, respectively. This architecture effectively utilizes the high-density ILVs for efficient on-chip data transfer and effectively overcomes the von Neumann bottleneck. The key contribution of this work lies in verifying the feasibility of such an M3D architecture, including fabrication process, functional demonstration, and performance evaluation.

To address your comment, we have systematically compared this work with the abovementioned references in **Table R3** below.

Reference	RRAM devices		Integration style
	Resistive switching layer	Function	
[2]	HfO ₂	CIM	2D
[3]	HfO ₂	Memory	M3D
[4]	HfO ₂	CIM	M3D

This work	HfO ₂ , Ta ₂ O ₅ (2 different types)	CIM, TCAM, Buffer	M3D
-----------	--	-------------------------	-----

Table R3. Comparison of this work with suggested literature.

Changes in the manuscript:

In page 4: In this work, for the first time, we report an M3D chip of RRAM-based hybrid memory architecture for efficient implementation of one-shot learning, **where the RRAM-based CIM, TCAM and buffer arrays serve different functions** as illustrated in **Figure 1a**.

Comment#3:

While the paper claims to showcase a completely integrated system, it fails to demonstrate its complete performance in a comprehensive manner. Here, HfAlOx-based RRAM is used for analog operations in CIM layer owing to the gradual resistive switching and multi-level programming capability. However, in dc-IV of HfAlOx-based RRAM shown in Fig.2e demonstrate low MW (<10x). The paper lacks a clear explanation of how it achieves 32 levels with HfAlOx, with such a low memory window (MW). Moreover, HfAlOx-based memristors are filamentary and intrinsically stochastic in nature, exhibiting high variability. Although endurance and cumulative behavior are shown for 2-levels of conductance states, more multi-level endurance data is needed to determine whether the 32 distinct levels are maintained without disturbance.

Response:

Thanks for the comment. First of all, the requirement on the memory window (MW) for CIM is different from that for digital memory application. A MW of about 10× is commonly reported for CIM application in literature (e.g., Peng Yao, et al., Nat. Commun., 2017, doi: 10.1038/ncomms15199), as there is usually a tradeoff between the MW and switching linearity (e.g., Wei Wu, et al., IEEE EDL, 2017, doi: 10.1109/LED.2017.2719161). Also, there is no direct evidence that suggests a correlation between the MW and multi-level programming capability in HfO₂-based RRAM devices. These two parameters are independent of each other. The multi-level programming capability of RRAM primarily depends on the materials it is based on. In a previous work (Peng Yao, et al., Nature, 577, 641-646, 2020, doi: 10.1038/s41586-020-1942-4), we have already demonstrated that with a MW close to 10, it is feasible to achieve 32 levels of programming. In this work, the 32-level programming in **Fig. 2g** is done by standard write-verify scheme, as illustrated in **Supplementary Fig. 2**.

Secondly, the endurance test is primarily conducted to evaluate the reliability of RRAM after subjecting to a specified number of cycles of electrical stress. Typically, only the maximum electrical stress is tested to drive the RRAM into the HRS and LRS,

rather than multiple intermediate resistance states (MRS). We shall point out that the endurance test itself is extremely time consuming, where the array-level endurance test with HRS and LRS only already took ~ 5 hours. Therefore, endurance test with multiple resistance states is seldomly reported in literature because the even more time-consuming write-verify operations are required for the programming of MRS. We performed endurance tests using the recommended method (Mario Lanza, et al., Adv. Elec. Mater., 2019, doi:10.1002/aelm.201800143), and the results are displayed in **Supplementary Fig.3**.

Additionally, due to the error-tolerant capability of neural networks, introducing certain degree of error or noise during the programming of the weight matrix would not significantly impact the overall accuracy. For instance, **Figure 2k** displays noticeable discrepancies between the predicted inner product outputs and the expected values. Similar findings have also been reported in the literature (e.g., Weier Wan, et al., Nature, 2022, doi: 10.1038/s41586-022-04992-8). Nevertheless, it is proved that such discrepancies do not significantly affect the accuracy and can be further reduced through algorithm and hardware co-optimizations.

Supplementary Fig.2. Programming of analog 1T1R cells in the 2nd layer of CIM array. (a) Illustration of the overall programming process. (b) Illustration of the set operation. (c) Illustration of the reset operation.

Supplementary Fig. 3. Array-level endurance and retention measurement on the analog RRAM-based CIM array. (a) Endurance test of the CIM array with 1024 1T1R cells. In each cycle, the 1T1R cells were mapped to the LRS and HRS using a write-verify scheme. The read voltage was 0.2 V. (b) Retention test of 8 representative conductance states. For the retention test, 1T1R cells in the 1k-bit array were mapped to 8 representative conductance states with 128 cells for each state. After mapping, their conductance was continuously measured after every second, and the average conductance of 128 1T1R cells for each state is plotted. (c) Illustration of full-fledged endurance test using pulse operation and reading (Mario Lanza, et al., Adv. Elec. Mater., 2019, doi:10.1002/aelm.201800143). The waveform of bit line voltage is plotted with the key parameters listed in the table below. (d) Full-fledged endurance test result of a representative 1T1R cell.

Changes in the manuscript:

In page 7:

Meanwhile, the HfAlO_x-based RRAM exhibited a more gradual resistive switching that enables excellent multi-level programming capability, which is **considered** favorable for CIM applications⁴⁴.

In reference:

44. Yao, P. *et al.* Face classification using electronic synapses. *Nat Commun* **8**, 15199 (2017).

Comment#4:

The paper does not provide sufficient specifications for the GPU used, particularly in terms of speed, area, and energy consumption. The M3D-LIME and GPU are compared only based on the energy and accuracy, where the paper also reports low accuracy in comparison to GPU, which lowers the impact of work. How do the M3D-LIME area and speed (execution time) compare with the GPU?

Response:

Thanks for your comment. Following your suggestion, we have made further clarifications on the comparison with GPU. In this work, we selected the Nvidia TESLA V100 as the GPU for comparison, which is based on the 16 nm technology node. Here we did not directly compare the area and speed of GPU and M3D-LIME chip (130nm) as their technology nodes are vastly different. Instead, we compare the energy efficiency as shown in the **Supplementary Table 4**. Here we adopted the reported energy efficiency value (100 GOPS/W) of NVIDIA Tesla V100 GPU, which has been widely used as a standard reference in many prior works (e.g., S. Ambrogio et al., *Nature* 2018; P. Yao et al., *Nature*, 2020.) for energy efficiency benchmark against GPU.

As to the slightly lower accuracy of M3D-LIME (96%) compared to GPU (98%), it is quite reasonable for CIM based on emerging memories (e.g., RRAM and PCM) to achieve lower accuracy than digital processors like GPU that computes with (double) floating point precision. This is mainly due to the non-ideal device characteristics and limited precision of emerging memories. The key advantage of RRAM-based CIM lies in the significantly higher energy efficiency than GPU at the cost of slightly lower but still acceptable accuracy. In addition, as this work focuses on the demonstration of a novel M3D-LIME architecture for one-shot learning, a more important and meaningful comparison is between M3D-LIME and 2D baseline, which has been comprehensively evaluated in **Fig. 4e** and **Supplementary Table 3**.

Changes in the manuscript:

In page 12: The benchmark results in **Figures 4d** and **4e** show that M3D-LIME could achieve $18.3\times$ higher energy efficiency than GPU (**Nvidia Tesla V100 as a commonly used reference for benchmark^{37,51}**) as well as a $2.73\times$ speedup than its 2D counterpart.

In Supplementary Table 4: Energy consumption evaluation for (a) TCAM, (b) CIM arrays, (c) buffer and (d) M3D-LIME as well as comparison of energy efficiency with GPU. #These values are obtained by the simulator XPEsim (Wenqiang Zhang, et al., DAC, 2019, doi: 10.1145/3316781.3317797). &Peripheral CMOS circuits, including

the ADCs, DACs, WL switch, and BL switch are all taken into consideration. * The reported energy efficiency value of the Nvidia Tesla V100 GPU (Stefano Ambrogio, et al., Nature, 2018, doi: 10.1038/s41586-018-0180-5), which has been extensively utilized as a standard reference in numerous prior studies is included for comparison in the energy efficiency benchmark against GPUs. **The Nvidia Tesla V100 GPU is fabricated at 12 nm technology node.**

Comment#5:

The energy consumption of the CIM array is obtained using the XPESim simulator (as mentioned in the Supplementary section). The parameters used for simulation are not reported. How do the energy values computed via simulations match the actual hardware?

Response:

Thanks for the comment. The XPESim simulator is an important open-source platform (see <https://github.com/thuime/XPESim> and Wenqiang Zhang, et al., DAC, 2019, doi: 10.1145/3316781.3317797) for us to evaluate the performance of CIM architecture. It has been carefully calibrated with extensive experimental data from our CIM hardware (Yuyi Liu, et al., IEEE TED, 2021, doi: 10.1109/TED.2021.3069746), and has been used for benchmark in several published works (e.g., Peng Yao, et al., Nature, 2020, doi: 10.1038/s41586-020-1942-4; Han Zhao, et al., Nature communications, 2023, doi: 10.1038/s41467-023-38021-7). In this work, the energy consumption of the CIM array was evaluated as follows:

First, the input parameters for the XPESim simulator were determined. Based on the characterization results of CIM array, the RRAM conductance was in the range of 4-40 μ S. For the specific task, four CIM array sizes were required: 9×64 , 9×128 , 6217×8 , and 8×128 . Also, 130 nm technology node was used here.

Using these parameters, the power consumption calculations were performed using XPESim. This evaluation can be divided into three components:

- 1) The power consumption of the RRAM crossbar array was primarily determined by the array size and RRAM conductance range. The array size and RRAM conductance range were input parameters, while the weight distribution was simulated using XPESim. For instance, an 8×128 array with a conductance range of 4-40 μ S resulted in a power consumption value of 32.6 pJ for the RRAM crossbar array.
- 2) The power consumption of the CIM array control circuitry and drivers, including WL/SL switches and MUX, was mainly determined by the technology node as well as the number of inputs and outputs of the CIM array. XPESim utilized DC synthesis and layout analysis to calculate the power consumption based on these

parameters. For example, in an 8×128 array with a technology node of 130 nm, the power consumption values were found to be 8.7 pJ for the WL switch, 3.3 pJ for the SL switch, and 11.1 pJ for the MUX.

- 3) The power consumption of the ADC was mainly determined by the technology node and the number of outputs. XPEsim referred to the appropriate references for the technology node of 130 nm (5.04-fJ/conversion-Step: Taimur Rabuske, et al., IEEE TVLSI, 2014, doi: 10.1109/TVLSI.2014.2337236) and calculated the ADC's power consumption based on the number of outputs. For instance, for an array with 128 outputs, the power consumption was found to be 0.17 nJ: $5.04 \text{ (fJ/conversion - setp)} \times 2^8 \text{ (bit)} \times 128 \text{ (outputs)} = 0.17 \text{ nJ}$.

Based on these calculations, **Supplementary Table 4b** presents the results. Since the power consumption of the second component was significantly lower than that of the third one, and both parts were implemented using Si CMOS circuits, they were combined as the power consumption of the "CMOS" part.

Energy Consumption of CIM Arrays [#]				
	9 * 64	9 * 128	6217 * 8	8 * 128
CMOS ^{&}	91.3 pJ	0.17 nJ	7.85 nJ	0.17 nJ
CIM array	18.3 pJ	36.7 pJ	7.36 nJ	32.6 pJ
Total	0.11 nJ	0.21 nJ	15.2 nJ	0.20 nJ

Supplementary Table 4b. Energy consumption evaluation for CIM arrays. [#]These values are obtained by the simulator XPEsim (Wenqiang Zhang, et al., DAC, 2019, doi: 10.1145/3316781.3317797). [&]Peripheral CMOS circuits, including the ADCs, DACs, WL switch, and BL switch are all taken into consideration.

Comment#6:

The details of the 2D counterpart of the hybrid system used for execution time comparison with M3D-LIME are missing. The details are necessary to understand the execution time benchmark provided in the supplementary section of the paper. The reported values are not intuitive, and additional explanations should be provided.

Response:

Thank you for pointing this out. To address this concern, we have now provided a dedicated section in the Supplementary Information to illustrate the 2D counterpart of the hybrid memory system which helps explain the execution time benchmark. In this work, the execution time is evaluated as follows:

To estimate the latency, the parameters used were listed, as shown in **Supplementary Table 3a**. The technology node used for the first layer of Si CMOS was 130 nm. Frequency of 200 MHz and bandwidth of 128 bits were assumed for the 2D data bus. The write delay of RRAM buffer (~50 ns) was assessed using the newly added experimental data shown in the **Supplementary Fig.7** below. The buffer read

delay was mainly determined by SA, and a typical value of 10 ns reported in literature (e.g., S. D. Spetalnick, et al., ISSCC, 2022, doi: 10.1109/ISSCC42614.2022.9731725; C.-C. Chou, et al., ISSCC, 2018, doi: 10.1109/ISSCC.2018.8310392) was used. The latency of CIM core was calculated by XPESim, considering the entire CIM array as a whole, from receiving data to outputting results, including the switching time of the array WL/SL switch and ADC based on the technology node, ADC parameters, and clock frequency. Binary quantization can be implemented using logic circuits, and its latency was roughly assumed to be one clock cycle.

Based on these parameters, we conducted pipeline planning for the tasks to evaluate the latencies of the 2D baseline (illustrated in **Supplementary Fig. 11**) and M3D-LIME. The difference between them is that 2D chip used a 2D bus for data transfer, thus limiting the number of MVM computations that can be performed in parallel by the CIM arrays, while the M3D-LIME communicated using fine-grained and high-density ILVs, whose bandwidth did not limit the number of arrays computing in parallel. Considering the specific neural network, we conducted pipeline planning and obtained the results shown in **Supplementary Figure 12**. The bus occupancy rate and the time of the array waiting for data were classified as the data transfer time to represent the communication efficiency for 2D baseline and M3D.

Supplementary Fig.7. Waveforms of set/reset pulse operations on 1T1R cell in the TCAM layer. (a) Waveform of the BL voltage and current during the set operation. The pulse width is ~ 50 ns. (b) Waveform of the BL voltage and current during the reset operation. The pulse width is also ~ 50 ns. Initially, a 0.15V pulse was used to measure the initial resistance (current) of the RRAM. Subsequently, a 50 ns pulse with a set voltage of 1.5V or a reset voltage of 3V was applied to operate the RRAM. Finally, a 0.15V pulse was used to measure the post-operation resistance (current) of the RRAM. The results demonstrated that RRAM can be successfully programmed by a 50 ns pulse, with a set voltage of 1.5 V and a reset voltage of 3 V.

Changes in the manuscript:

In page 12: Furthermore, the execution time and energy consumption were also evaluated to manifest the advantages of M3D-LIME compared to GPU and 2D chip architecture **which is illustrated in the Supplementary Fig.11.**

Supplementary Fig. 11. Architecture of the 2D baseline chip. In the 2D chip architecture, the CIM core, the TCAM, and the cache are realized in the same manner as in the M3D-LIME. However, the key difference lies in their way of on-chip data transfer, as the M3D-LIME uses high-bandwidth ILVs while the 2D counterpart employs a bus. Specifically, the CIM core is realized by HfAlO_x-based analog RRAM array and Si CMOS control circuits, while the TCAM and cache employ CNTFETs and Ta₂O₅-based binary RRAMs. In the benchmark evaluation, the bus has a bandwidth of 128 bits and a frequency of 200 MHz. Due to the limited bandwidth of the bus, substantial data transfer latency is incurred when performing parallel computing in multiple CIM arrays, which acts as a bottleneck for the overall system performance.

Comment#7:

The paper's Supplementary section only includes figures and captions, with no accompanying explanations or descriptions, making it difficult for readers to understand the study's results and implications.

Response:

Thank you very much for the comment. To improve the readability of the Supplementary section, we have now added three Supplementary notes to give more detailed explanations and descriptions.

Changes in the manuscript:

Add three Supplementary notes:

Supplementary Note 1: Experimental method of results shown in Figure 2i-k.

In this experiment, we employed a multi-channel memory testing system in conjunction with the peripheral circuitry integrated on-chip with the CIM array (as depicted in **Supplementary Fig.1**) to demonstrate the calculation of MVM. The steps

are summarized below:

To begin, we first mapped the weight matrix into the CIM array by programming. Initially, we determined the mapping relationship between the weights and the RRAM conductance values. For this demonstration, we set the memory window of the RRAM in the range of 0.4 to 40 μS , divided evenly into 16 conductance states (equivalent to 4 bits). Each weight in the weight matrix was quantized to 4 bits and then mapped to the respective RRAM conductance state, serving as the target value for programming the CIM array.

Subsequently, using the test system, we programmed each RRAM in the CIM array based on its target conductance using the standard write-verify method. We input a 7-bit binary address through the WL address input port of the CIM array to select the WL of the RRAM to be programmed. By controlling the voltage applied to the gate of the transistor connected to the selected WL through the WL reference voltage port, while keeping other WLs at a constant voltage of 0 V to ensure the connected transistors were turned off and prevent cross-talk, we could select one row of RRAMs (a total of 8) via WLs and then select the individual RRAMs one by one using the BLs (Bit Lines) orthogonal to the WLs.

For programming each 1T1R cell as shown in **Supplementary Fig.2**, we defined an interval based on the target conductance and the programming margin. The objective was to program the RRAM conductance within this interval, for which we applied a series of set and reset pulses. After each pulse, we read the conductance of the RRAM. If it exceeded the upper limit of the interval, we performed a reset operation in the next cycle. Conversely, if it fell below the lower limit, we performed a set operation in the next cycle. This process continued until the RRAM conductance was within the interval or the maximum number of cycles was reached, indicating the completion of programming. We then proceeded to program the next RRAM. The set and reset pulses had a width of 50 ns, and their voltages ranged from 1 V to 3 V, with WL reference voltages of 1 V to 3 V for set pulses and 5 V for reset pulses. For reading the conductance, we applied a WL reference voltage of 5 V and a BL voltage of 0.15 V.

Once all the RRAMs were programmed, we completed the mapping of the weight matrix to the CIM array. By controlling the WL address input and BL, we read the results of the entire programmed array, as depicted in **Figure 2i**. The difference between this result and the target conductance value is shown in **Figure 2j**.

After that, we proceeded to input a series of vectors into the CIM array to demonstrate the MVM calculation. These input vectors followed a Gaussian distribution with a mean of 0 and a variance of 1, with each element of the vectors representing the elements of the matrix. These vectors were multiplied by the target conductance matrix, yielding the MVM result vector, where each element represented

the expected inner product output.

Subsequently, by controlling the WL address and BL voltage, we performed 100 read operations on the CIM array. These operations were subject to read noise and write errors, resulting in fluctuating conductance matrices compared to the target conductance matrix. We multiplied the input vectors with these matrices, obtaining the predicted MVM results from the CIM array. In order to compare the difference between the predicted MVM results and the theoretical results, we considered the predicted inner product output as a function of its theoretical value, i.e., the expected inner product output. To mitigate the influence of output amplitude on the results, the data was normalized as plotted in **Figure 2k**, which demonstrates the capability of the CIM array in performing the MVM calculation.

Supplementary Note 2: Experimental method of results shown in Figure 3c.

Instruments utilized for this experiment included the Keysight B1500 and B1530 Semiconductor Analyzers, Keysight B2201 Switch Matrix, Tektronix MS054 Oscilloscope, and Probe Card System. To demonstrate the calculation of Hamming distance, we followed the steps outlined below:

Firstly, for each 2T2R cell in the TCAM arrays, we programmed all the RRAMs according to the scheme described in **Supplementary Table 1**. Specifically, we sequentially programmed each RRAM device by applying suitable voltages to the SELs, while turning off the driving transistors of the unselected RRAM (typical gate voltage of 0 V) and setting the gate voltage of the transistor driving the selected RRAM (typical gate voltage of -2.5 V). Next, we applied appropriate operating voltages to the ML and SL for DC programming of the devices. For the set operation, we applied 0V to the ML and -2 V to the SL. For the reset operation, we applied -3 V to the ML and 0V to the SL. After the set/reset operation, we then performed DC readout of the programmed RRAMs by applying -0.15 V to the ML, 0 V to the SL, -5 V to the SEL of the selected RRAM, and 0V to the SEL of the unselected RRAMs. If the resistance of the selected RRAM did not meet the requirements (typical values were around 30 k Ω for LRS and above 5 M Ω for HRS), we then applied a series of 50 ns pulses to continue programming the RRAM. After each 50 ns pulse operation, we read the resistance value of the RRAM. If it did not meet the requirements, we increased the pulse operating voltage for the next cycle until the resistance value fell within the target range to complete the programming. We repeated this process for each device until every unit in the 5 \times 1 TCAM array was correctly programmed. Through these operations, we successfully stored the template vector in the TCAM array. For the testing in Figure 3c, a '00000' vector is stored.

After that, search vectors were inputted through the SELs to calculate the Hamming

distance. In this experiment, we first connected the ML to the oscilloscope and set the source electrode of the pre-charging transistor to 1 V. Then, based on each bit of the search vector, we defined the on and off states of each transistor during the search process according to the scheme outlined in **Supplementary Table 1**. For the transistors in the off state, their connected SELs were grounded with SL, resulting in a gate voltage of 0 throughout the search process. For the SELs of transistors in the on state during the search process and the gate electrode of the pre-charging transistor, they were respectively connected to two pulse sources. Initially, the pre-charging transistor was turned on (typically with a gate voltage of -4 V) to charge the ML. After a certain period of time (typically 1 ms), the pre-charging transistor was turned off (typically with a gate voltage of 1 V), followed by a delay (typically 1 μ s). Then, by applying a -5 V pulse to the corresponding SELs, we turned on the corresponding transistors and triggered the oscilloscope to record the discharge waveform of the ML. The relevant pulses were applied through a set of PMUs of Keysight B1530, enabling good synchronization of the signals. By implementing search vectors with different Hamming distances from the stored template and recording the results on the waveforms, we obtained the results depicted in **Figure 3c**.

Supplementary Note 3: Testing method for Supplementary Fig.8.

We built a testing system with Keysight B1500 and B1530 for TCAM-related testing. Initially, we programmed the 2T2R TCAM cells using the configuration detailed in **Supplementary Table 1**. This involved applying distinct voltages to the gate terminals of the two CNTFETs to turn on the one driving the selected RRAM while keeping the other one driving the unselected RRAM turned off ($V_{GS}=0$). We also applied the appropriate voltages to the ML and SL in order to program each RRAM in the 2T2R cell. The typical voltages for set operation were $V_{GS}=-3$ V and $V_{ML}=-2$ V, whereas $V_{GS}=-5$ V and $V_{SL}=-3$ V for reset. Following the programming of the 2T2R cells, we conducted the 1-bit search operation on them using the configuration outlined in **Supplementary Table 1**. This included applying a voltage of 0.15 V to the ML and appropriate V_{GS} voltages to the two CNTFETs, then reading the current and calculating the resistance. In cases where the search data did not match the stored template, the CNTFET driving the RRAM in LRS opened and the CNTFET driving the RRAM in HRS closed, leading to a lower resistance known as mismatch resistance. Conversely, if the search data matched the stored template, the CNTFET driving the HRS opened and the CNTFET driving the LRS closed, resulting in a larger resistance known as match resistance. The ratio between the match and mismatch resistances serves as a key parameter for RRAM-based TCAM as it determines the length of the WL, or the search length.

Reviewer #3

General comment:

This is a very solid work in which the authors fabricate a monolithically 3D integrated small-sized chip that comprises a Si CMOS layer with two additional stacked BEOL-compatible layers respectively composed of analog RRAM for CIM core computing and binary RRAM for buffering and TCAM purposes. The authors report noteworthy results in terms of functional integration of the different functionalities on the same chip. The work represents an advancement as compared to existing literature and, in general, the claims are well supported. The adopted methodology is solid and the data are well presented, with a good amount of details being outlined. The paper is well written and enjoyable to read. However, a general limitation of the work is that the performance results of the classification task are just simulated since the authors fabricated a small-sized version of the chip having a single 1 by 5 TCAM array. In this respect, extra care must be taken when giving off numbers coming from the system-level simulations. While this reviewer understands that the fabrication of a full-sized chip is probably beyond the scope of this work, it is advisable to take care of some crucial aspects, outlined in the following.

Response:

We would like to express our deep appreciation to the reviewer for taking the time to review our manuscript and providing valuable feedback. We greatly appreciate the acknowledgment of the manuscript's novelty and the insightful comments made by the reviewer. We have carefully considered the concerns raised by the reviewer and have taken them into account in the revision. We have conducted additional experiments and provided more details in the revised manuscript. The point-by-point response to your technical comments are as follows.

Comment#1:

The most important point to be considered is that the delay and power consumption introduced by the data interface that, according to what the authors state includes DACs and ADCs (and possibly additional components and circuitry) is completely ignored. These elements can easily constitute the delay and energy bottleneck. This makes the estimated energy efficiency figure of merit inaccurate and the direct comparison with the GPU to be taken with a grain of salt, at best, or actually meaningless, at worst. A similar matter holds true for the latency benchmark vs. the 2D chip: if the delay ends up being dominated by the data interface, then the advantage of the M3D approach (at least in terms of speedup) vanishes. In this respect, either more

accurate calculations needs to be done or at least these elements should be properly indicated in the paper.

Response:

Thanks for your comment. In the estimation of power consumption and latency, we did take into account all the peripheral circuits including ADCs, DACs, WL switch, and BL switch, which were all grouped as Si CMOS circuits when calculating the power consumption. Similarly, when calculating the latency, we referred the entire CIM array as the CIM core, and the overall time from when the data is first inputted to when the computation is completed and the final output is provided by the ADCs was defined as the CIM core latency.

As the reviewer correctly anticipated, the ADCs indeed contributed substantially to the power consumption of the entire CIM chip, it still exhibits considerable energy efficiency advantage compared to GPU. Furthermore, despite the limitation imposed by the ADCs on the speed of the CIM core, our evaluation has revealed high data throughput when multiple arrays compute in parallel. Consequently, in the traditional 2D architecture, the number of arrays computing in parallel is constrained by the data bandwidth. In comparison, with the new M3D architecture utilizing high-bandwidth interlayer vias (ILVs), the parallelism can be significantly enhanced, resulting in improved computational speed for complex tasks.

To clarify this point, we have provided the following explanations in the revision.

Changes in the manuscript:

-In Supplementary Table 3: Supplementary Table 3. Parameters (a) and results (b) of the benchmark of execution time. **The CIM core latency includes the time from when the data is first inputted to when the computation is completed and the final output is provided by the ADCs.*

-In Supplementary Table 4:

a

Energy Consumption of TCAM	
$R_{\text{match}} / R_{\text{mismatch}}$	30 k Ω / 200 M Ω (1 bit)
V_{ML}	1 V
Average Search Energy	$[(1 \text{ V})^2 / 30 \text{ k}\Omega + (1 \text{ V})^2 / 200 \text{ M}\Omega] / 2 * 100 \text{ ns} / \text{bit} = 1.67 \text{ pJ/bit}$

b

Energy Consumption of CIM Arrays [#]				
	9 * 64	9 * 128	6217 * 8	8 * 128
CMOS ^{&}	91.3 pJ	0.17 nJ	7.85 nJ	0.17 nJ
CIM array	18.3 pJ	36.7 pJ	7.36 nJ	32.6 pJ
Total	0.11 nJ	0.21 nJ	15.2 nJ	0.20 nJ

c

Energy Consumption of Buffer	
LRS / HRS	30 k Ω / 200 M Ω
Write Voltage	3 V (Reset) / 1.5 V (Set)
Average Write Energy	$[(3 \text{ V})^2 / 30 \text{ k}\Omega + (1.5 \text{ V})^2 / 200 \text{ M}\Omega] / 2 * 50 \text{ ns} / \text{bit} = 7.50 \text{ pJ/bit}$
Read Voltage	0.15 V
Average Read Energy	$[(0.15 \text{ V})^2 / 30 \text{ k}\Omega + (0.15 \text{ V})^2 / 200 \text{ M}\Omega] / 2 * 10 \text{ ns} / \text{bit} = 3.75 \text{ fJ/bit}$

d

Energy Consumption of M3D-LIME	
Conv1	$0.11 \text{ nJ} * 4 * 196 + (64 + 9) * (3.75 \text{ fJ/bit} + 7.50 \text{ pJ/bit}) * 4 * 196 = 0.52 \text{ }\mu\text{J}$
Conv2	$0.21 \text{ nJ} * 64 * 196 + (128 + 9) * (3.75 \text{ fJ/bit} + 7.50 \text{ pJ/bit}) * 64 * 196 = 15.5 \text{ }\mu\text{J}$
FC1	$15.2 \text{ nJ} + (6217 + 8) * (3.75 \text{ fJ/bit} + 7.50 \text{ pJ/bit}) = 61.9 \text{ nJ}$
FC2	$0.20 \text{ nJ} + (128 + 8) * (3.75 \text{ fJ/bit} + 7.50 \text{ pJ/bit}) = 1.22 \text{ nJ}$
TCAM	$1.67 \text{ pJ/bit} * 5 * 128 = 1.07 \text{ nJ}$
Total	16.1 μJ

e

Energy Efficiency Comparison		
	M3D-LIME	GPU [*]
Average Latency	9860 ns	-
Energy Consumption	16.3 μJ	-
Energy Efficiency	29.9 MOP / 16.3 $\mu\text{J} = 1.83 \text{ TOPs}^{-1}\text{W}^{-1}$	30.4 TOPs ⁻¹ / 300 W = 0.1 TOPs ⁻¹ W ⁻¹

Supplementary Table 4. Energy efficiency benchmark. Energy consumption evaluation for (a) TCAM, (b) CIM arrays, (c) buffer and (d) M3D-LIME as well as comparison of energy efficiency with GPU. [#]These values are obtained by the simulator XPEsim (Wenqiang Zhang, et al., DAC, 2019, doi: 10.1145/3316781.3317797). [&]Peripheral CMOS circuits, including the ADCs, DACs, WL switch, and BL switch are all taken into consideration. ^{*}The reported energy efficiency value of the Nvidia Tesla V100 GPU (Stefano Ambrogio, et al., Nature, 2018, doi: 10.1038/s41586-018-0180-5),

which has been extensively utilized as a standard reference in numerous prior studies is included for comparison in the energy efficiency benchmark against GPUs. The Nvidia Tesla V100 GPU is fabricated at 12 nm technology node.

Comment#2:

In supplementary table 4 the authors assume a 9 MOhm resistance for the mismatch case, which looks largely overestimated given the performance outlined in Fig. S5b (a value around 200 MOhm would be more representative). The authors should adopt a more realistic value, even if this does not change drastically the average write and read energies. Then, they assume that 1T1R buffer cells can be written in 50 ns with a single 3V (reset) / 1.5 V (set) pulse and read with a 0.15 V 10 ns pulse. However, this is not demonstrated in the paper - the authors are encouraged to show that achieving these performance levels is possible, providing a statistically significant validation. This is critical since the energy performance of the system largely depends on that of the buffer; given the numbers in supplementary table 4.

Response:

Thank you to the reviewer for providing insightful suggestions on improving the performance benchmark.

First, following your suggestion, we have adjusted the mismatch resistance of the TCAM and the HRS of the RRAM buffer to a logarithmic average value of 200 M Ω (Previously **Fig. S5b** shows a wide distribution from 9 M Ω to about 1000 M Ω). Using this number, we recalculated the power consumption of M3D-LIME in **Supplementary Table 4**, which is actually lower than previously calculated due to the increase of HRS.

Secondly, we have conducted additional pulse tests to analyze the operating voltage of the buffer, and the results have been added as **Supplementary Fig. 7**. Initially, a 0.15V pulse was used to measure the initial resistance of the RRAM. Subsequently, a 50 ns pulse with a set voltage of 1.5V or a reset voltage of 3V was applied to operate the RRAM. Finally, a 0.15V pulse was used to measure the post-operation resistance of the RRAM. The current waveform in the test results indicated successful transition of the RRAM between HRS and LRS following the set and reset operations. Additionally, the voltage waveform demonstrated that the pulse width and voltage of the set and reset operations met the assumptions of the energy consumption benchmark.

Furthermore, the actual data reading speed in the buffer depends on the sense amplifier (SA). We set this parameter to its typical value of 10 ns (Samuel D. Spetalnick, et al., ISSCC, 2022, doi: 10.1109/ISSCC42614.2022.9731725; Chung-Cheng Chou, et al., ISSCC, 2018, doi: 10.1109/ISSCC.2018.8310392) with a read voltage of 0.15 V.

The above new data and analysis has been added in the revised Supplementary Information. Additionally, in response to your Comment #7 later, we have added a histogram of the RRAM operating voltage during the endurance test, which further supports the performance evaluation of RRAM buffer.

Changes in the manuscript:

-In Supplementary Table 4:

a

Energy Consumption of TCAM	
$R_{\text{match}} / R_{\text{mismatch}}$	30 k Ω / 200 M Ω (1 bit)
V_{ML}	1 V
Average Search Energy	$[(1 \text{ V})^2 / 30 \text{ k}\Omega + (1 \text{ V})^2 / 200 \text{ M}\Omega] / 2 * 100 \text{ ns} / \text{bit} = 1.67 \text{ pJ/bit}$

b

Energy Consumption of CIM Arrays [#]				
	9 * 64	9 * 128	6217 * 8	8 * 128
CMOS [#]	91.3 pJ	0.17 nJ	7.85 nJ	0.17 nJ
CIM array	18.3 pJ	36.7 pJ	7.36 nJ	32.6 pJ
Total	0.11 nJ	0.21 nJ	15.2 nJ	0.20 nJ

c

Energy Consumption of Buffer	
LRS / HRS	30 k Ω / 200 M Ω
Write Voltage	3 V (Reset) / 1.5 V (Set)
Average Write Energy	$[(3 \text{ V})^2 / 30 \text{ k}\Omega + (1.5 \text{ V})^2 / 200 \text{ M}\Omega] / 2 * 50 \text{ ns} / \text{bit} = 7.50 \text{ pJ/bit}$
Read Voltage	0.15 V
Average Read Energy	$[(0.15 \text{ V})^2 / 30 \text{ k}\Omega + (0.15 \text{ V})^2 / 200 \text{ M}\Omega] / 2 * 10 \text{ ns} / \text{bit} = 3.75 \text{ fJ/bit}$

d

Energy Consumption of M3D-LIME	
Conv1	$0.11 \text{ nJ} * 4 * 196 + (64 + 9) * (3.75 \text{ fJ/bit} + 7.50 \text{ pJ/bit}) * 4 * 196 = 0.52 \mu\text{J}$
Conv2	$0.21 \text{ nJ} * 64 * 196 + (128 + 9) * (3.75 \text{ fJ/bit} + 7.50 \text{ pJ/bit}) * 64 * 196 = 15.5 \mu\text{J}$
FC1	$15.2 \text{ nJ} + (6217 + 8) * (3.75 \text{ fJ/bit} + 7.50 \text{ pJ/bit}) = 61.9 \text{ nJ}$
FC2	$0.20 \text{ nJ} + (128 + 8) * (3.75 \text{ fJ/bit} + 7.50 \text{ pJ/bit}) = 1.22 \text{ nJ}$
TCAM	$1.67 \text{ pJ/bit} * 5 * 128 = 1.07 \text{ nJ}$
Total	16.1 μJ

e

Energy Efficiency Comparison		
	M3D-LIME	GPU [*]
Average Latency	9860 ns	-
Energy Consumption	16.3 μJ	-
Energy Efficiency	29.9 MOP / 16.3 $\mu\text{J} = 1.83 \text{ TOPs}^{-1}\text{W}^{-1}$	30.4 TOPs ⁻¹ / 300 W = 0.1 TOPs ⁻¹ W ⁻¹

Supplementary Table 4. Energy efficiency benchmark. Energy consumption

evaluation for (a) TCAM, (b) CIM arrays, (c) buffer and (d) M3D-LIME as well as comparison of energy efficiency with GPU. #These values are obtained by the simulator XPEsim (Wenqiang Zhang, et al., DAC, 2019, doi: 10.1145/3316781.3317797). &Peripheral CMOS circuits, including the ADCs, DACs, WL switch, and BL switch are all taken into consideration. *The reported energy efficiency value of the Nvidia Tesla V100 GPU (Stefano Ambrogio, et al., Nature, 2018, doi: 10.1038/s41586-018-0180-5), which has been extensively utilized as a standard reference in numerous prior studies is included for comparison in the energy efficiency benchmark against GPUs. The Nvidia Tesla V100 GPU is fabricated at 12 nm technology node.

Add a new supplementary figure:

Supplementary Fig.7. Waveforms of set/reset pulse operations on 1T1R cell in the TCAM layer. (a) Waveform of the BL voltage and current during the set operation. The pulse width is ~50ns. (b) Waveform of the BL voltage and current during the reset operation. The pulse width is also ~50ns. Initially, a 0.15V pulse was used to measure the initial resistance (current) of the RRAM. Subsequently, a 50 ns pulse with a set voltage of 1.5V or a reset voltage of 3V was applied to operate the RRAM. Finally, a 0.15V pulse was used to measure the post-operation resistance (current) of the RRAM. The results demonstrated that RRAM can be successfully programmed by a 50 ns pulse, with a set voltage of 1.5 V and a reset voltage of 3 V.

Comment#3:

- Line 124: *It would be ideal to provide a reference for the Omninglot dataset.*

Response:

Thanks for your kind suggestion. We have now cited the Omninglot dataset in our revised manuscript as follows: Lake, B. M., Salakhutdinov, R., and Tenenbaum, J. B. (2015). Human-level concept learning through probabilistic program induction. *Science*, 350(6266), 1332-1338. doi:10.1126/science.aab3050.

Changes in the manuscript:

In page 5: System-level simulations indicated that the M3D-LIME achieved a GPU-equivalent accuracy of up to 96% on the Omniglot dataset²², while exhibiting 18.3×higher energy efficiency than GPU and 2.73× faster speed than its 2D counterpart.

In reference:

22. Lake, B. M., Salakhutdinov, R. & Tenenbaum, J. B. Human-level concept learning through probabilistic program induction. *Science* (1979) 350, 1332–1338 (2015).

Comment#4:

- Line 176: Please specify the set and reset pulse amplitude.

Response:

Thanks for the comment. The set and reset voltages are 1.6 V and 2.6 V, respectively.

Changes in the manuscript:

In page 7:

Figure 2f shows the analog resistive switching characteristics with good linearity and symmetry, where 20 1T1R cells were measured under a series of set and reset pulses with a width of 50 ns, ~~and the read voltage was 0.2 V.~~ The set, reset and read voltages are 1.6 V, 2.6 V and 0.2 V, respectively.

Comment#5:

- Line 181: It is advisable to show the full-fledged endurance plot of 1M cycles (i.e., through all cycles, not just using 1 point per decade).

Response:

We appreciate the reviewer's advice and have optimized the endurance test according to the recommended approach in the literature (Mario Lanza, et al., *Adv. Elec. Mater.*, 2019, doi:10.1002/aelm.201800143). We shall mention that the endurance test, especially on the array-level is extremely time consuming. Therefore, there have been different ways to evaluate the endurance of RRAM in literature. Two new subfigures have been included in **Supplementary Fig. 3**. The first figure provides a concise depiction of our endurance measurement method, which include read pulses in between write pulses to enable fast testing. The figure also includes a table that outlines the relevant parameters. The second figure presents the results of the full-fledged endurance testing using this method. Since **Supplementary Fig. 3a** has already displayed the statistical results for 1k devices, and conducting full-fledged endurance testing is time-consuming, we chose to only present the test results on a representative device in this instance, as shown in **Supplementary Fig. 3d**.

Changes in the manuscript:

-Added two new subfigures to Supplementary Fig. 3:

Supplementary Fig. 3. Array-level endurance and retention measurement on the analog RRAM-based CIM array. (a) Endurance test of the CIM array with 1024 1T1R cells. In each cycle, the 1T1R cells were mapped to the LRS and HRS using a write-verify scheme. The read voltage was 0.2 V. (b) Retention test of 8 representative conductance states. For the retention test, 1T1R cells in the 1k-bit array were mapped to 8 representative conductance states with 128 cells for each state. After mapping, their conductance was continuously measured after every second, and the average conductance of 128 1T1R cells for each state is plotted. (c) Illustration of full-fledged endurance test using pulse operation and reading (Mario Lanza, et al., Adv. Elec. Mater., 2019, doi:10.1002/aelm.201800143). The waveform of bit line voltage is plotted with the key parameters listed in the table below. (d) Full-fledged endurance test result of a representative 1T1R cell.

Comment#6:

- Line 191: Please give more details as far as the normalized vector is concerned. How is this represented in Fig. 2k? How do the authors represent as a single number the results of a MVM operation? Does this actually average out possible significant errors

in the individual entries of the output vector? Can the authors quantify the actual errors achieved at the single entry level?

Response:

Thank you for the comment regarding the demonstration of MVM operation in **Figure 2k**. In this demonstration, we first mapped a weight matrix to the RRAM array, where the conductance of the RRAM cells corresponds to the magnitude of the weight matrix elements. To accommodate the limited precision of RRAM devices, we quantized these weights into 4 bits. The corresponding mapping results are displayed in **Figure 2i**, while the associated mapping error is presented in **Figure 2j**.

Next, we generated a series of 8-bit vectors with each element following Gaussian distributions. These vectors were then inputted into the RRAM array, where the MVM operation was performed based on the principles of Kirchhoff's current law and Ohm's law. This operation yielded a series of 128-bit vectors as the output. **Figure 2k** plots all the elements of these output vectors in accordance with their expected output values.

In addition, due to the strong correlation between the absolute values of the output elements and the statistical characteristics of the weight matrix elements, we employed a normalization method to effectively calculate the relative computational errors. We extend our gratitude to the reviewer for bringing to our attention the issue with our previous description. While we did demonstrate the MVM, the results shown in **Figure 2k**, which were plotted to better showcase the computational error, should be more accurately referred to as "Inner product results", which has been corrected in the revision. To clarify this point, we have provided further explanations on the MVM demonstration in the revised manuscript.

Changes in the manuscript:

In page 7: To demonstrate this process on the fabricated 1k-bit CIM array, an 8×128 weight matrix used in the feature extraction later was first mapped on the CIM array as shown in **Figure 2i**, and the corresponding mapping error is shown in **Figure 2j**. ~~The conductance values of RRAM cells were proportional to the magnitude of the elements in the weight matrix, which were quantized into 4 bits. After the weight mapping, a series of vectors were input to the CIM array, and Figure 2. (k) plots the normalized calculation results of MVM using the mapped array versus the theoretically expected results.~~ ~~After the weight mapping, a series of 8-bit vectors, whose elements followed Gaussian distributions, were input into the RRAM array for performing MVM operations. By applying the Kirchhoff's current law and Ohm's law, the RRAM array completed the MVM calculations and output a series of 128-bit vectors. All the elements of these vectors represented the inner products were normalized and plotted against their theoretically expected results in **Figure 2k**.~~ The linear fitting suggests a good consistency between them with a small R-square of 0.96. This result confirms the

feasibility of performing MVM operations efficiently on the analog RRAM array for CIM.

In Fig.2:

Figure 2. Characterizations of analog RRAM-based CIM layer with M3D stacked RRAM buffer. (a) Optical image of the 2nd layer of CIM array before the fabrication of the 3rd layer. Scale bar: 60 μm . Inset: zoom-in view of the CIM array, scale bar: 10 μm . (b) Optical image (left) and SEM image (right) of the 2nd layer after the fabrication of the 3rd layer. Scale bar: 5 μm . Inset: zoom-in view of the fabricated RRAM buffer array, scale bar: 1 μm . (c) and (d) Cross-sectional TEM images of the HfAlO_x-based analog RRAM and Ta₂O₅-based digital RRAM, respectively, scale bar: 20 nm. (e) DC I-V measurements of HfAlO_x-based RRAM with analog resistive switching characteristics and Ta₂O₅-based digital RRAM with a large HRS/LRS ratio. (f) Analog

resistive switching characteristics of 20 1T1R cells in the CIM array under a series of set and reset pulses. Gray lines are the raw data, and blue line plots the average conductance of 20 devices measured. (g) Cumulative probability distribution of the CIM array with 32 equally distributed conductance states, showing the programming capability of 5 bits per cell. (h) Illustration of performing MVM operation on the CIM array. (i) Mapped conductance G_{mapped} and (j) corresponding error G_{error} after mapping a matrix on the 1k-bit CIM array. (k) Calculation result of MVM using the mapped array shown in (i) versus the theoretically expected result. The blue line plots the linear fitting with a small R-square of 0.96.

Comment#7:

- Line 239: Similar to what stated previously, it is advisable to show the full-fledged endurance plot of 50k cycles. Also, in Fig. S4 it is clear that the memory window can go as low as 10. The authors are encouraged to show some statistical endurance and retention results on several devices.

Response:

Thank you very much for the comment on the endurance and retention tests.

Regarding the endurance test, we attempted two methods. We first tried the first method as depicted in Supplementary **Fig. 3c-d** in response to Comment #5, by adding a read pulse after each set/reset pulse. The corresponding results are shown in **Figure R3** below. For the Ta₂O₅-based binary RRAM, it had a large on/off ratio and the HRS (in the range of 9 M Ω ~1000 M Ω) was over the current reading limit in this mode. Therefore, the readout HRS resistance was around 200 k Ω as shown in **Fig. R2**. For sanity check, we also used DC reading and the measured resistance of HRS was above 10 M Ω , which represented the actual HRS value. Because of this issue, the full-fledged endurance test in **Fig. R2** shows a memory window of about 10.

To avoid this problem, we then opted for the second method to characterize the endurance of Ta₂O₅-based binary RRAM, as shown in **Figure R4**. We first combined pulse operation with DC reading to provide full-fledged Endurance test results for 1000 cycles, demonstrating the successful switching between HRS and LRS in each cycle. However, performing a DC reading after every cycle was extremely time-consuming, taking approximately 1 hour for 1000 cycles. As a result, we adopted a scheme where a read was conducted after multiple cycles (10 times per decade), which allowed us to measure the endurance up to 10⁶ cycles. This method is also recommended in literature as a tradeoff between measurement time and test cycles with meaningful precision

(Mario Lanza, et al., ACS NANO, 2021, doi:10.1021/acsnano.1c06980). Additionally, we have performed this endurance test on 10 devices to present the statistical characteristics in Supplementary Fig. S6d.

For the statistical retention test, we also included the retention characteristics of 25 devices with HRS and 25 devices with LRS.

Based on the new experimental results above, we have made revisions to the corresponding figures and manuscript accordingly.

Figure R3. Full-fledged endurance test result of a typical 1-CNTFET-1-RRAM cell.

Figure R4. Illustration of endurance test using pulse operation and DC reading.

Changes in the manuscript:

-In Supplementary Fig. 6:

Supplementary Fig. 6. Characterizations of 1T1R half-cell in the TCAM layer. (a) Cross-sectional TEM image of a 1T1R half-cell, scale bar: 200 nm. **(b)** Endurance test and **(c)** retention test (baking at 120 °C) of a typical 1T1R half-cell, exhibiting a large HRS/LRS ratio. **(d)** Histogram of set and reset voltages during the endurance test mentioned in **(b)**. The endurance test involved the measurement on 10 1T1R half-cells, while the retention test was performed on 25 1T1R half-cells with LRS and 25 1T1R half-cells with HRS. The CNTFETs used to drive the Ta₂O₅-based RRAM cells have a channel size of W/L=20 μm/2 μm.

Comment#8:

- Line 241: In Fig. S5a the fact that there is no crosstalk is not evident to this reviewer. The authors should better convey their message perhaps using better data representation or a more dedicated approach. The legend in Fig. S5d can be misleading although the data are clear.

Response:

Thanks for the comment. Firstly, let's explain why **Fig. S5a** (now **Supplementary Fig. 8** in the revision) suggests no crosstalk. In **Fig. S5a**, we conducted consecutive DC operations on two RRAMs in a 2T2R TCAM cell, following the sequence 1-2-3-4 as indicated in the **Figure R5**. Assuming that the CNTFET did not act as a good selector and crosstalk occurs, if one RRAM (RRAM2 in this case) was set to LRS during the first operation, there would be a significant leakage current during the subsequent

operation due to crosstalk at a low BL voltage. However, we observed that the current during the subsequent operation in the figure was minimal (suggesting RRAM1 was still in HRS) when a low BL voltage was applied. Therefore, we interpret this as an indication of the absence of crosstalk. We appreciate the reviewer for bringing up the concern that this reasoning may be overly complex and potentially misleading. Moreover, since it is not a critical conclusion, we have decided to remove this statement.

Secondly, we have made a correction on the legend of **Fig. S5d**. We appreciate the reminder from the reviewer.

Figure R5. Illustration of 4-step DC operations on two RRAMs in a 2T2R TCAM cell.

Changes in the manuscript:

-In Supplementary Fig. 8:

Supplementary Fig. 8. Characterizations of 2T2R TCAM cells in the TCAM layer.

(a) DC I-V curves of two 1T1R half-cells in a 2T2R TCAM cell, ~~showing no crosstalk when operating one of them.~~ (b) Distribution of discharging resistance for 100 2T2R cells, showing a large match/mismatch resistance ratio $>300\times$. Results of the read disturb test are shown in (c) and (d): (c) Search '1' when '1' is stored in the TCAM cell. (d) Search '1' when '0' is stored in the TCAM cell. In the search operation, 100-ns pulses with a voltage of 1 V are applied to the TE (ML). The CNTFETs used in the 2T2R TCAM cells have a channel size of $20\ \mu\text{m}/2\ \mu\text{m}$.

REVIEWER COMMENTS

Reviewer #1 (Remarks to the Author):

The authors improved significantly the paper and answered most of this reviewer's comments. Still, a few supplementary figures could help the understanding of the paper.

1. A single-figure schematic showing the overall experiment with a 2D visualization of the chip (partially shown in SI Fig.1 and Fig.3b) is missing. I think would be useful to see an overall schematic to really understand easily what is off-chip and on-chip (it can be highlighted with different colors similarly to what was done for SI Fig. 1) and also what is on-chip in different layers (again different colors would help)

2. A schematic and vision for an actual architecture is missing. The authors replied in Comment 11 (rev1) but only described what the chip looks like. My questions are: how big of a workload could you put in a scale-up chip? Do you need multiple "cores"? how do the "cores" communicate? Is a multichip architecture even possible? Of course, a detailed answer is out of the scope of this work, but having a general vision in the paper could help in understanding how and if this proposal would actually scale up.

Reviewer #2 (Remarks to the Author):

The manuscript demonstrates a monolithic three-dimensional integration of hybrid memory architecture (M3D LIME) based on resistive memories to implement one-shot learning.

The work demonstrated here is an improvement from the previously published IEDM paper (titled "Monolithic 3D Integration of Logic, Memory and Computing-In-Memory for One-Shot Learning" published in 2021 from the same group). While the work would be of interest to the community, it doesn't qualify as the first demonstration of integrating multiple types of RRAMs. The demonstration of the integrated hybrid memory structure presented in IEDM should take precedence.

The reviewer recommends the authors to modify the Abstract (page 2, line 29-32) and Introduction (page 4, line 108-111) accordingly.

All other technical questions are addressed.

Reviewer #3 (Remarks to the Author):

The paper largely improved as compared to the previous version. This reviewer thinks the authors did a satisfactory job in addressing this reviewer's comments. Nevertheless, in this reviewer's opinion, a few points need to be taken care of before the paper can be published.

- In Fig. S1, it looks like SLs are shared but in the text, when describing figure 2h, the authors state "the current outputs on the SLs represent the MVM results". This is correct if SLs are not shared, otherwise MVM operation would result in a scalar, not a vector. The authors should either update Fig. S1 or let the reader better understand why SLs are shared.

- In Fig. 2j, perhaps showing the percentage or relative deviation instead of the absolute deviation would be better.

- In the supplementary material, TCAM search time is considered to be 100 ns for execution time and energy calculations. However, in Fig. 3f it looks like a few microseconds are necessary to have a dependable assessment of the hamming distance. In addition, how large is - realistically - the pre-charging time of the ML in the actual integrated chip? How much does it affect the overall TCAM execution time?

- In the execution time benchmarking it is unclear how and if max pooling time is included.

- In Supplementary Table 2, Conv layer 1 is reported to be $3 \times 3 \times 64 \times 3$ and FC1 6272×8 in size. However in Supplementary Table 3b Conv layer 1 is reported to be simulated as $4 \times 9 \times 64$ arrays and FC1 as a single 6217×8 array. What is the origin of this discrepancy?

- In the supplementary material, in the table of content "Supplementary Note 3: Testing method for Supplementary Fig.7." is reported, but in the text the related chapter is titled "Supplementary Note 3: Testing method for Supplementary Fig.8.". The authors may want to fix this.

- In Supplementary Table 4, when the authors state that "Peripheral CMOS circuits, including the ADCs, DACs, WL switch, and BL switch are all taken into consideration" it is advisable to include a few words describing the way in which these contributions are considered (referring to the literature and reporting some information as the authors did in the response letter).

Response Letter to reviewers' Comments

We sincerely appreciate the reviewers' insightful and constructive comments. We have carried out additional experiments and revised the manuscript according to the reviewers' suggestions. Below are the point-by-point responses to each comment. All the changes to the manuscript are marked in red.

Reviewer #1

General comment:

The authors improved significantly the paper and answered most of this reviewer's comments. Still, a few supplementary figures could help the understanding of the paper.

Response:

We appreciate the valuable time the reviewer took to review our paper and provide constructive feedback. In response to the reviewer's technical comments, we have provided point-to-point responses below and included additional information in the revised supplementary information. We hope you find the revised manuscript satisfactory.

Comment#1:

A single-figure schematic showing the overall experiment with a 2D visualization of the chip (partially shown in SI Fig.1 and Fig.3b) is missing. I think would be useful to see an overall schematic to really understand easily what is off-chip and on-chip (it can be highlighted with different colors similarly to what was done for SI Fig. 1) and also what is on-chip in different layers (again different colors would help).

Response:

We appreciate the reviewer's valuable feedback and suggestions. In response to the reviewer's comment, we have revised Supplementary Fig. 1 to include additional subfigures depicting the CIM and TCAM arrays as well as the signal transmissions and the testing systems. It can now provide a more comprehensive overview of our experimental setup.

Changes in the manuscript:

Revised supplementary figure S1:

Supplementary Fig. 1. Overall experimental setup of the M3D-LIME. (a) Schematic diagram of the 1Kb CIM array used for implementing MVM calculations. The peripheral circuits are implemented using Si CMOS logic in the first layer of M3D-LIME chip. #Usually it is preferred not to connect all the SLs together in order to perform fully parallel MVM calculation. However, in practical circuit design, multiple or all the SLs could be connected together to share one ADC, which helps save the hardware cost as well as reduce the number of test pads and simplify testing. In this case, the selector transistors of a specific row can be activated through the WLS, allowing the MVM calculation on one SL at one time. This enables sequential MVM calculation. (b) Illustration of the signal transmissions between the CIM array, TCAM array and the testing system. (c) Schematic diagram of the 5×1 TCAM array used for implementing Hamming distance calculations. The CNTFETs and Ta₂O₅-based RRAMs of the TCAM array are fabricated in the 3rd layer.

Comment#2:

A schematic and vision for an actual architecture is missing. The authors replied in Comment 11 (rev1) but only described what the chip looks like. My questions are: how big of a workload could you put in a scale-up chip? Do you need multiple “cores”? how do the “cores” communicate? Is a multichip architecture even possible? Of course, a detailed answer is out of the scope of this work, but having a general vision in the paper could help in understanding how and if this proposal would actually scale up.

Response:

Thanks for further clarifying your comment. As illustrated in the newly added **Supplementary Fig. 13**, a scale-up M3D-LIME chip would consist of multiple CIM cores, one or more TCAM arrays (depending on the application scenario), as well as the underneath Si CMOS control logic circuits. Each "core" represents a processing element (PE) that contains one CIM array and associated on-chip buffer. Multiple PEs could further form a tile. The number of CIM PEs and tiles integrated on the chip depends on the specific application as well as other factors such as the technology node and chip area. In addition to the on-chip buffers, network-on-chip (NoC) based on routers can be employed to facilitate communication between those cores. In this way, large-scale memory-augmented neural networks can be implemented efficiently on M3D-LIME.

Changes in the manuscript:

In page 13 (Conclusion): **As illustrated in Supplementary Fig. 13, a scale-up M3D-LIME chip could monolithically integrate multiple CIM arrays, the associated on-chip buffers, and one or more TCAM arrays, on top of Si CMOS logic circuits to efficiently implement large-scale MANNs.** Our work demonstrates the tremendous potential of M3D with hybrid memory architecture for future data-intensive AI and HPC applications.

Add a new supplementary figure:

Supplementary Fig. 13. Illustration of a scale-up M3D-LIME chip. (a) Illustration of a processing unit (PE) that contains one CIM core, associated on-chip buffer and CMOS logic. (b) Illustration of a tile, containing multiple PEs, one TCAM and one CMOS control logic. (c) Illustration of a TCAM. The scale-up M3D-LIME chip could monolithically integrate multiple CIM arrays, the associated on-chip buffers, and one or more TCAM arrays, on top of Si CMOS logic circuits to efficiently implement large-scale MANNs. Each core represents a processing element (PE) that contains one CIM array, associated on-chip buffer and CMOS logic. Multiple PEs, one TCAM and one CMOS control logic could further form a tile. The number of PEs and tiles integrated on the chip depends on the specific application as well as other factors such as the technology node and chip area. In addition to the on-chip buffers, network-on-chip (NoC) based on routers can be employed to facilitate communication between those cores.

Reviewer #2

General comment:

The manuscript demonstrates a monolithic three-dimensional integration of hybrid memory architecture (M3D LIME) based on resistive memories to implement one-shot learning. The work demonstrated here is an improvement from the previously published IEDM paper (titled "Monolithic 3D Integration of Logic, Memory and Computing-In-Memory for One-Shot Learning" published in 2021 from the same group). While the work would be of interest to the community, it doesn't qualify as the first demonstration of integrating multiple types of RRAMs. The demonstration of the integrated hybrid memory structure presented in IEDM should take precedence.

The reviewer recommends the authors to modify the Abstract (page 2, line 29-32) and Introduction (page 4, line 108-111) accordingly.

All other technical questions are addressed.

Response:

We sincerely appreciate the reviewer's positive feedback on our revised manuscript, and we are glad to see you are satisfied with our responses to your technical questions. To address your last concern on the first demonstration of integrating multiple types of RRAMs, we have revised the corresponding statements in the Abstract (Page 2, lines 29-32) and Introduction (Page 4, lines 108-111) sections as well as in the Conclusions section.

Changes in the manuscript:

In page 2 (Abstract): In this work, we report ~~the first~~ a monolithic three-dimensional integration (M3D) of hybrid memory architecture based on resistive random-access memory (RRAM), named M3D-LIME, to efficiently implement one-shot learning.

In page 4 (Introduction): In this work, ~~for the first time~~, we report an M3D chip of RRAM-based hybrid memory architecture for efficient implementation of one-shot learning, where the RRAM-based CIM, TCAM and buffer arrays serve different functions as illustrated in **Figure 1a**.

In page 13 (Conclusion): In sum, ~~for the first time~~, we have designed and fabricated a M3D-LIME chip with a hybrid memory architecture of RRAM-based CIM, buffer, and TCAM for efficiently implementing one-shot learning.

Reviewer #3

General comment:

The paper largely improved as compared to the previous version. This reviewer thinks the authors did a satisfactory job in addressing this reviewer's comments. Nevertheless, in this reviewer's opinion, a few points need to be taken care of before the paper can be published.

Response:

We sincerely appreciate the reviewer's positive feedback on our revised manuscript. In response to your further concerns, we have provided point-to-point responses as follows.

Comment#1:

In Fig. S1, it looks like SLs are shared but in the text, when describing figure 2h, the authors state "the current outputs on the SLs represent the MVM results". This is correct if SLs are not shared, otherwise MVM operation would result in a scalar, not a vector. The authors should either update Fig. S1 or let the reader better understand why SLs are shared.

Response:

Thank you for the valuable comment. You are absolutely correct that it is preferred not to connect all the SLs together in order to perform fully parallel MVM calculation. However, in practical circuit design, multiple or all the SLs could be connected together to share one ADC, which helps save the hardware cost as well as reduce the number of test pads and simplify testing. In this case, the selector transistors of a specific row can be activated through the WLs, allowing the MVM calculation on one SL at one time. This enables sequential MVM calculation.

In response to your comment, we have modified Fig. S1 and clarified the SL connections in the figure caption.

Changes in the manuscript:

In Supplementary Fig.1:

Supplementary Fig. 1. Overall experimental setup of the M3D-LIME. (a) Schematic diagram of the 1Kb CIM array used for implementing MVM calculations. The peripheral circuits are implemented using Si CMOS logic in the first layer of M3D-LIME chip. #Usually it is preferred not to connect all the SLs together in order to perform fully parallel MVM calculation. However, in practical circuit design, multiple or all the SLs could be connected together to share one ADC, which helps save the hardware cost as well as reduce the number of test pads and simplify testing. In this case, the selector transistors of a specific row can be activated through the WLS, allowing the MVM calculation on one SL at one time. This enables sequential MVM calculation. (b) Illustration of the signal transmissions between the CIM array, TCAM array and the testing system. (c) Schematic diagram of the 5×1 TCAM array used for implementing Hamming distance calculations. The CNTFETs and Ta₂O₅-based RRAMs of the TCAM array are fabricated in the 3rd layer.

Comment#2:

In Fig. 2j, perhaps showing the percentage or relative deviation instead of the absolute deviation would be better.

Response:

Thank you for your comment. We have revised **Figure 2j** in the manuscript accordingly.

Changes in the manuscript:

Figure 2j:

Comment#3:

In the supplementary material, TCAM search time is considered to be 100 ns for execution time and energy calculations. However, in Fig. 3f it looks like a few microseconds are necessary to have a dependable assessment of the hamming distance. In addition, how large is - realistically - the pre-charging time of the ML in the actual integrated chip? How much does it affect the overall TCAM execution time?

Response:

Thank you for your comment. Firstly, in **Fig. 3f**, we experimentally fabricated and measured a TCAM array with a search line length of 5. It was measured that the discharging time constant τ for 5 mismatched bits ($N_{\text{mis}} = 5$) is about 3 μs . As plotted in **Fig. 3e**, the reciprocal of discharging time constant τ exhibits a linear dependence on N_{mis} . In the simulation of one-shot learning task, the simulated TCAM array has a search line length of 128. In this case, we can roughly extrapolate that the discharging time constant for $N_{\text{mis}} = 128$ would be $\tau = 3 \mu\text{s} / 128 \times 5 = 117 \text{ ns}$, which is close to the assumed TCAM search time of $\sim 100 \text{ ns}$ in **Supplementary Table 3**. We shall point out that the measured discharging time constant is mainly limited by the parasitic capacitance in our testing system (including a switching matrix and probe card), which is significantly larger than the on-chip parasitic capacitance of MLs. Therefore, we are confident that a TCAM search time shorter than 100 ns can be reasonably achieved if we eliminate the undesired parasitic capacitance and also shrink the channel length of CNTFET (currently 2 μm) that helps reduce the ML pre-charging time.

To clarify this point, we have added the above discussion in the revised Supplementary Information.

Changes in the manuscript:

-In Supplementary Table 3:

a

Parameters of Execution Time Benchmark		
	M3D-LIME	2D Baseline
Technology Node	130 nm	
Bandwidth of Bus	-	128 bit
Clock	200 MHz	200 MHz
Size of the Input Image	28 * 28	
Buffer write latency	50 ns	
Buffer read latency	10 ns	
CIM core latency*	50 ns	
Binary Quantization latency	5 ns	
TCAM latency	100 ns [#]	

b

Execution Time Benchmark ^a				
		Array number	Array size	Latency
Conv1	M3D-LIME	4	9 * 64	9860 ns
	2D Baseline			11758.4 ns
Conv2	M3D-LIME	64	9 * 128	9860 ns
	2D Baseline			26920 ns
FC1	M3D-LIME	1	6217 * 8	110 ns
	2D Baseline			2070 ns
FC2	M3D-LIME	1	8 * 128	110 ns
	2D Baseline			115 ns
TCAM	M3D-LIME	1	5 * 128	115 ns
	2D Baseline			
Average / pipeline latency of M3D-LIME				9860 ns
Average / pipeline latency of 2D Baseline				26920 ns

Supplementary Table 3. Parameters (a) and results (b) of the benchmark of execution time. *The CIM core latency includes the time from when the data is first inputted to when the computation is completed and the final output is provided by the ADCs. [#]The TCAM latency of 100ns for a search length of 128 is roughly extrapolated from the experimental data in Figure 3c (where the discharging time for $N_{mis} = 5$ with a search length of 5 is about 3 μ s) using the formula: $\tau = 3 \mu\text{s}/128 \times 5 = 117 \text{ ns}$. We shall point out that the measured discharging time constant is mainly limited by the parasitic capacitance in our testing system (including a switching matrix and probe card). The TCAM search time can be significantly reduced if we eliminate the undesired parasitic capacitance and also shrink the channel length of CNTFET that helps reduce the ML

pre-charging time. &The max pooling time is taken into consideration, but it actually does not increase the overall computation time, since it can be implemented using high-speed digital logic circuits and performed in parallel with the MVM calculation in the next stage.

Comment#4:

In the execution time benchmarking it is unclear how and if max pooling time is included.

Response:

Thanks for your comment. The max pooling time is considered in the execution time benchmarking, but it actually does not increase the overall computation time. For example, for Conv1, with a size of $1 \times 64 \times 3 \times 3$ (as corrected in the response to **Comment #5**), for an input image size of 28×28 , it requires four 9×64 arrays to compute in parallel for $28 \times 28 / 4$ times (as mentioned in the response to **Comment #5**). After each computation, the results are stored in the buffer, and during the time it takes to complete the next MVM calculation, max pooling can be performed in parallel. Moreover, max pooling can be implemented using high-speed digital logic circuits, so its computation time ($5 \text{ clock cycles} \times 1/200 \text{ MHz} = 25 \text{ ns}$ for 3×3 max pooling by Bin Zhao, et al., IECON, 2020, doi: 10.1109/IECON43393.2020.9254452) usually does not exceed the computation time of the CIM array (50 ns). As a result, it is not expected to affect the pipeline delay.

To clarify this point, we have added the above discussion in the revised Supplementary Information.

Changes in the manuscript:

In Supplementary Table 3:

a

Parameters of Execution Time Benchmark		
	M3D-LIME	2D Baseline
Technology Node	130 nm	
Bandwidth of Bus	-	128 bit
Clock	200 MHz	200 MHz
Size of the Input Image	28 * 28	
Buffer write latency	50 ns	
Buffer read latency	10 ns	
CIM core latency*	50 ns	
Binary Quantization latency	5 ns	
TCAM latency	100 ns [#]	

b

Execution Time Benchmark ^{&}				
		Array number	Array size	Latency
Conv1	M3D-LIME	4	9 * 64	9860 ns
	2D Baseline			11758.4 ns
Conv2	M3D-LIME	64	9 * 128	9860 ns
	2D Baseline			26920 ns
FC1	M3D-LIME	1	6217 * 8	110 ns
	2D Baseline			2070 ns
FC2	M3D-LIME	1	8 * 128	110 ns
	2D Baseline			115 ns
TCAM	M3D-LIME	1	5 * 128	115 ns
	2D Baseline			
Average / pipeline latency of M3D-LIME				9860 ns
Average / pipeline latency of 2D Baseline				26920 ns

Supplementary Table 3. Parameters (a) and results (b) of the benchmark of execution time. *The CIM core latency includes the time from when the data is first inputted to when the computation is completed and the final output is provided by the ADCs. [#]The TCAM latency of 100ns for a search length of 128 is roughly extrapolated from the experimental data in Figure 3c (where the discharging time for $N_{\text{mis}} = 5$ with a search length of 5 is about 3 μs) using the formula: $\tau = 3 \mu\text{s}/128 \times 5 = 117 \text{ ns}$. We shall point out that the measured discharging time constant is mainly limited by the parasitic capacitance in our testing system (including a switching matrix and probe card). The TCAM search time can be significantly reduced if we eliminate the undesired parasitic capacitance and also shrink the channel length of CNTFET that helps reduce the ML pre-charging time. [&]The max pooling time is taken into consideration, but it actually does not increase the overall computation time, since it can be implemented using high-

speed digital logic circuits and performed in parallel with the MVM calculation in the next stage.

Comment#5:

In Supplementary Table 2, Conv layer 1 is reported to be $3 \times 3 \times 64 \times 3$ and FC1 6272×8 in size. However, in Supplementary Table 3b Conv layer 1 is reported to be simulated as 4 9×64 arrays and FC1 as a single 6217×8 array. What is the origin of this discrepancy?

Response:

Thanks for pointing out this discrepancy. Yes, the size of the Conv1 layer should be $1 \times 64 \times 3 \times 3$, where 1 (instead of 3) represents the number of input image channels. The size of the CIM array required for a fully connected layer is determined by the network size. Therefore, for a fully connected layer of size 6217×8 (FC1), a CIM array of the same size, 6217×8 , is needed. For the convolutional layer, such as $1 \times 64 \times 3 \times 3$ (Conv1), the kernel size is 3×3 , which means it processes 9 input pixels at a time. Here 64 represents the number of output channels. Therefore, the CIM array size used is 9×64 . Since the input image size is 28×28 (**Supplementary Table 3b**), convolution must be performed 28×28 times. To expedite computation, we employ a strategy of copying the weight matrix 4 times and performing parallel computation using 4 CIM arrays (**Supplementary Table 3b**), which also leads to 4 times higher power consumption.

Furthermore, the output of Conv1 is $28 \times 28 \times 64$, which then undergoes max pooling, resulting in a size of $14 \times 14 \times 64$ before being input to Conv2. The Conv2 layer has an input channel number of 64 and requires 9×128 CIM arrays to perform convolutions $14 \times 14 \times 64$ times. The power consumption for Conv2 can be calculated using the same method as Conv1.

In response to your comment, we have made the following modifications in the benchmark:

1. Correct the erroneous Conv1 size.
2. Revise the statement on the calculation in **Supplementary Table 4d**.
3. Add explanations for the specific data in the caption of **Supplementary Table 4**.

We believe that the revised supplementary information can now provide a clear explanation on the relationship between array size, calculation count, and neural network dimensions.

a

Parameters of Execution Time Benchmark		
	M3D-LIME	2D Baseline
Technology Node	130 nm	
Bandwidth of Bus	-	128 bit
Clock	200 MHz	200 MHz
Size of the Input Image	28 * 28	
Buffer write latency	50 ns	
Buffer read latency	10 ns	
CIM core latency*	50 ns	
Binary Quantization latency	5 ns	
TCAM latency	100 ns [#]	

b

Execution Time Benchmark [‡]				
		Array number	Array size	Latency
Conv1	M3D-LIME	4	9 * 64	9860 ns
	2D Baseline			11758.4 ns
Conv2	M3D-LIME	64	9 * 128	9860 ns
	2D Baseline			26920 ns
FC1	M3D-LIME	1	6217 * 8	110 ns
	2D Baseline			2070 ns
FC2	M3D-LIME	1	8 * 128	110 ns
	2D Baseline			115 ns
TCAM	M3D-LIME	1	5 * 128	115 ns
	2D Baseline			
Average / pipeline latency of M3D-LIME				9860 ns
Average / pipeline latency of 2D Baseline				26920 ns

Supplementary Table 3. Parameters (a) and results (b) of the benchmark of execution time. *The CIM core latency includes the time from when the data is first inputted to when the computation is completed and the final output is provided by the ADCs. [#]The TCAM latency of 100ns for a search length of 128 is roughly extrapolated from the experimental data in Figure 3c (where the discharging time for $N_{\text{mis}} = 5$ with a search length of 5 is about 3 μs) using the formula: $\tau = 3 \mu\text{s}/128 \times 5 = 117 \text{ ns}$. We shall point out that the measured discharging time constant is mainly limited by the parasitic capacitance in our testing system (including a switching matrix and probe card). The TCAM search time can be significantly reduced if we eliminate the undesired parasitic capacitance and also shrink the channel length of CNTFET that helps reduce the ML pre-charging time. [&]The max pooling time is taken into consideration, but it actually does not increase the overall computation time, since it can be implemented using high-

speed digital logic circuits and performed in parallel with the MVM calculation in the next stage.

Changes in the manuscript:

-In Supplementary Table 2:

Convolutional Layer 1		Convolutional Layer 2		Fully Connected Layers and TCAM	
Conv	1×64×3×3	Conv	64×128×3×3	FC1	6272×8
BN	-	BN	-	FC2	8×128
ReLU	-	ReLU	-	Binary Quantization	-
MaxPooling	2×2	MaxPooling	2×2	TCAM	128×5
Dropout	-	Dropout	-		

Supplementary Table 2. Key parameters of the MANN implemented in this work for one-shot learning.

a

Energy Consumption of TCAM	
$R_{\text{match}} / R_{\text{mismatch}}$	30 k Ω / 200 M Ω (1 bit)
V_{ML}	1 V
Average Search Energy	$[(1 \text{ V})^2 / 30 \text{ k}\Omega + (1 \text{ V})^2 / 200 \text{ M}\Omega] / 2 * 100 \text{ ns} / \text{bit} = 1.67 \text{ pJ/bit}$

b

Energy Consumption of CIM Arrays [#]				
	9 * 64	9 * 128	6217 * 8	8 * 128
CMOS ^{&}	91.3 pJ	0.17 nJ	7.85 nJ	0.17 nJ
CIM array	18.3 pJ	36.7 pJ	7.36 nJ	32.6 pJ
Total	0.11 nJ	0.21 nJ	15.2 nJ	0.20 nJ

c

Energy Consumption of Buffer	
LRS / HRS	30 k Ω / 200 M Ω
Write Voltage	3 V (Reset) / 1.5 V (Set)
Average Write Energy	$[(3 \text{ V})^2 / 30 \text{ k}\Omega + (1.5 \text{ V})^2 / 200 \text{ M}\Omega] / 2 * 50 \text{ ns} / \text{bit} = 7.50 \text{ pJ/bit}$
Read Voltage	0.15 V
Average Read Energy	$[(0.15 \text{ V})^2 / 30 \text{ k}\Omega + (0.15 \text{ V})^2 / 200 \text{ M}\Omega] / 2 * 10 \text{ ns} / \text{bit} = 3.75 \text{ fJ/bit}$

d

Energy Consumption of M3D-LIME	
Conv1	$0.11 \text{ nJ} * 28 * 28 + (64 + 9) * (3.75 \text{ fJ/bit} + 7.50 \text{ pJ/bit}) * 28 * 28 = 0.52 \text{ }\mu\text{J}^*$
Conv2	$0.21 \text{ nJ} * 64 * 14 * 14 + (128 + 9) * (3.75 \text{ fJ/bit} + 7.50 \text{ pJ/bit}) * 64 * 14 * 14 = 15.5 \text{ }\mu\text{J}$
FC1	$15.2 \text{ nJ} + (6217 + 8) * (3.75 \text{ fJ/bit} + 7.50 \text{ pJ/bit}) = 61.9 \text{ nJ}$
FC2	$0.20 \text{ nJ} + (128 + 8) * (3.75 \text{ fJ/bit} + 7.50 \text{ pJ/bit}) = 1.22 \text{ nJ}$
TCAM	$1.67 \text{ pJ/bit} * 5 * 128 = 1.07 \text{ nJ}$
Total	16.1 μJ

e

Energy Efficiency Comparison		
	M3D-LIME	GPU ^{##}
Average Latency	9860 ns	-
Energy Consumption	16.3 μJ	-
Energy Efficiency	29.9 MOP / 16.3 $\mu\text{J} = 1.83 \text{ TOPs}^{-1}\text{W}^{-1}$	30.4 TOPs ⁻¹ / 300 W = 0.1 TOPs ⁻¹ W ⁻¹

Supplementary Table 4. Energy efficiency benchmark. Energy consumption evaluation for (a) TCAM, (b) CIM arrays, (c) buffer and (d) M3D-LIME as well as comparison of energy efficiency with GPU. [#]These values are obtained by the simulator XPesim (Wenqiang Zhang, et al., DAC, 2019, doi: 10.1145/3316781.3317797). [&]Peripheral CMOS circuits, including the ADCs, DACs, WL switch, and BL switch are all taken into consideration. ^{*}Input image size is 28×28 . ^{##}The reported energy

efficiency value of the Nvidia Tesla V100 GPU (Stefano Ambrogio, et al., Nature, 2018, doi: 10.1038/s41586-018-0180-5), which has been extensively utilized as a standard reference in numerous prior studies is included for comparison in the energy efficiency benchmark against GPUs. The Nvidia Tesla V100 GPU is fabricated at 12 nm technology node.

Comment#6:

In the supplementary material, in the table of content "Supplementary Note 3: Testing method for Supplementary Fig.7." is reported, but in the text the related chapter is titled "Supplementary Note 3: Testing method for Supplementary Fig.8.". The authors may want to fix this.

Response:

Thank you very much for pointing this out. We have corrected the typo in the revised supplementary information.

Changes in the manuscript:

-In Table of Contents (Supplementary Information): Supplementary Note 3: Testing method for Supplementary Fig.7.8.

Comment#7:

In Supplementary Table 4, when the authors state that "Peripheral CMOS circuits, including the ADCs, DACs, WL switch, and BL switch are all taken into consideration" it is advisable to include a few words describing the way in which these contributions are considered (referring to the literature and reporting some information as the authors did in the response letter).

Response:

Thank you for the reviewer's kind suggestion. In response to your comment, we have now added a supplementary note in the revised supplementary information to clarify the evaluation method for the energy consumption of the CIM array.

Changes in the manuscript:

Add a new Supplementary Note:

Supplementary Note 4: Evaluation method for the energy consumption of the CIM array (as presented in Supplementary Table 4.b).

In this work, we employed the XPESim simulator, an open-source platform (available at <https://github.com/thuime/XPESim>) introduced by our previous work (Wenqiang Zhang, et al., DAC, 2019, doi: 10.1145/3316781.3317797), to assess the energy consumption of the CIM array. Here the energy consumption of the CIM array was evaluated as follows:

First, the input parameters for the XPESim simulator were determined. Our measurement results of the CIM array indicated that the RRAM conductance was in the range of 4-40 μ S. For the specific task, four CIM array sizes were employed: 9 \times 64, 9 \times 128, 6217 \times 8, and 8 \times 128. Also, 130 nm technology node was used here.

Using these parameters, the power consumption calculations were performed using XPESim. This evaluation can be divided into three components:

- 1) The power consumption of the RRAM array was primarily determined by the array size and RRAM conductance range, which were taken as input parameters, while the weight distribution was simulated using XPESim. For instance, an 8 \times 128 array with a conductance range of 4-40 μ S resulted in a power consumption value of 32.6 pJ for the RRAM array.
- 2) The power consumption of the peripheral circuitry and drivers, including DACs, WL/SL switches and MUX, was mainly determined by the technology node as well as the number of inputs and outputs of the CIM array. XPESim utilized DC synthesis and layout analysis to calculate the power consumption based on these parameters. For example, in an 8 \times 128 array with a technology node of 130 nm, the power consumption values were estimated to be 8.7 pJ for the WL switch, 3.3 pJ for the SL switch, and 11.1 pJ for the MUX.
- 3) The power consumption of ADCs was mainly determined by the technology node and the number of outputs. XPESim referred to the appropriate references using the technology node of 130 nm (e.g., 5.04-fJ/conversion-Step: Taimur Rabuske, et al., IEEE TVLSI, 2014, doi: 10.1109/TVLSI.2014.2337236) and calculated the ADC's power consumption based on the number of outputs. For instance, for an array with 128 outputs, the power consumption was found to be 0.17 nJ:
5.04 (fJ/conversion – setp) \times 2⁸ (bits) \times 128 (outputs) = 0.17 nJ.

Finally, we summed up the power consumption of peripheral circuitry, drivers, and ADCs and recorded it under the label of “CMOS” in **Supplementary Table 4b**.

REVIEWERS' COMMENTS

Reviewer #1 (Remarks to the Author):

The authors addressed all the concerns raised by this reviewer.

Reviewer #3 (Remarks to the Author):

The authors have nicely addressed all of this reviewer's concerns - no further comments.

Response Letter to reviewers' Comments

We are delighted to see that both reviewers are satisfied with our revision and recommend publication of our work in Nature Communications without further technical comments. We sincerely appreciate the valuable time the reviewers have spent reviewing our manuscript and providing insightful comments and suggestions, which definitely help further improve the quality of our work.

Reviewer #1

General comment:

The authors addressed all the concerns raised by this reviewer.

Response:

We are very glad to see that you are satisfied with our revision. We sincerely thank you for the valuable time you have spent reviewing our manuscript and providing insightful comments to help significantly improve the quality of our work.

Reviewer #3

General comment:

The authors have nicely addressed all of this reviewer's concerns - no further comments.

Response:

We are very glad to see that you are satisfied with our revision. We sincerely thank you for the valuable time you have spent reviewing our manuscript and providing insightful comments to help significantly improve the quality of our work.